# Genome-wide association analysis of self-reported daytime sleepiness identifies 42 loci that suggest biological subtypes

Heming Wang ⓘ et al.[#]

Excessive daytime sleepiness (EDS) affects 10–20% of the population and is associated with substantial functional deficits. Here, we identify 42 loci for self-reported daytime sleepiness in GWAS of 452,071 individuals from the UK Biobank, with enrichment for genes expressed in brain tissues and in neuronal transmission pathways. We confirm the aggregate effect of a genetic risk score of 42 SNPs on daytime sleepiness in independent Scandinavian cohorts and on other sleep disorders (restless legs syndrome, insomnia) and sleep traits (duration, chronotype, accelerometer-derived sleep efficiency and daytime naps or inactivity). However, individual daytime sleepiness signals vary in their associations with objective short vs long sleep, and with markers of sleep continuity. The 42 sleepiness variants primarily cluster into two predominant composite biological subtypes - sleep propensity and sleep fragmentation. Shared genetic links are also seen with obesity, coronary heart disease, psychiatric diseases, cognitive traits and reproductive ageing.

---

Excessive daytime sleepiness (EDS) is a chief symptom of chronic insufficient sleep[1] as well as of several primary sleep disorders, such as sleep apnea, narcolepsy, and circadian rhythm disorders[2,3]. Several disease processes and medications also associate with prevalent and incident EDS[4–6]. EDS is estimated to contribute to risk for motor vehicle crashes, work-related accidents, and loss of productivity, highlighting its public health importance[7,8]. The clinical impact of EDS extends to a negative impact on cognition, behavior, and quality of life[9]. Therefore, sleep interventions often identify reduction in EDS as a chief goal. EDS is also associated with an increased risk for cardio-metabolic disorders, psychiatric problems, and mortality[6,10] through pathways that may be causal, bi-directional, or reflect pleiotropic effects.

While EDS occurs in a variety of settings associated with insufficient sleep, there is large inter-individual variability in levels of EDS that is not fully explained by sleep duration, sleep quality, or chronic disease[11]. Experimental studies have shown that there is also individual vulnerability to EDS following sleep restriction[11,12]. The heritability of daytime sleepiness is estimated to be between 0.37 and 0.48 in twin studies[13–15], 0.17 in family studies[16], and between 0.084 and 0.17 in GWAS[17,18], suggesting that genetic factors contribute to variation in sleepiness. Despite multiple candidate gene studies[19] and GWAS[17,20,21], including one from the first genetic release of the UK Biobank[18], few genome-wide significant genetic variants have been reported, likely reflecting the heterogeneous and multifactorial etiology of the phenotype and low statistical power.

Here, we extend our GWAS of self-reported daytime sleepiness to the full UK Biobank dataset[22] and identify multiple genetic variants grouping into different biological subtypes that associate with sleepiness. Bioinformatics analyses further highlight relevant biological processes and reveal shared genetic background with other diseases.

## Results

**Sample characteristics.** In the UK Biobank[22], 452,071 participants of European genetic ancestry self-reported the frequency of daytime sleepiness using the question: "How likely are you to doze off or fall asleep during the daytime when you don't mean to? (e.g.: when working, reading or driving)", with the answer categories "never" ($N = 347,285$), "sometimes" ($N = 92,794$), "often" ($N = 11,963$), or "all of the time" ($N = 29$). The severity of daytime sleepiness increased with older age, female sex, higher body mass index (BMI), various behavioral, social and environmental factors, and chronic diseases (Supplementary Table 1). Self-reported daytime sleepiness was positively but weakly correlated with self-reported insomnia symptoms, morning chronotype, ICD-10, or self-report of physician diagnosed sleep apnea and self-reported shorter and longer sleep duration, consistent with earlier reports or known clinical correlates[18] (Spearman correlation <0.2; Supplementary Table 1 and Supplementary Fig. 1). Self-reported daytime sleepiness was also weakly correlated with shorter sleep duration, lower sleep efficiency (indicating more time awake during the sleep period), and longer daytime inactivity duration estimated using a 7-day accelerometry in a subset ($N = 85,388$) of UK Biobank participants (Methods; Supplementary Table 2).

**GWAS, sensitivity, and replication analyses.** We performed a GWAS of self-reported daytime sleepiness treating the four categories as a continuous variable using a linear mixed regression model[23] adjusted for age, sex, genotyping array, ten principal components (PCs) of ancestry and genetic relatedness matrix, and identified 37 genome-wide significant loci ($P < 5 \times 10^{-8}$) (Fig. 1,

Supplementary Fig. 2, Supplementary Table 3). The most significant association was observed within the gene *KSR2*, a gene associated with multiple physiological pathways relevant to sleep and metabolism[24,25] (see Discussion). Additional novel loci were identified within or near genes with known actions on sleep–wake control regulation or that are associated with sleep disorders (e.g. *PLCL1* (ref. [26]), *GABRA2* (ref. [27]), *BTBD9* (ref. [28]), *HTR7* (ref. [29]), *RAI1* (ref. [30])), metabolic traits (e.g. *GCKR*[31], *SLC39A8* (ref. [32])), and psychiatric traits (e.g. *AGAP1* (ref. [33]), *CACNA1C*[34]). Regional association plots of genome-wide significant loci are shown in Supplementary Fig. 3. We identified 37 association signals driven by common lead variants with minor allele frequency 0.08–0.49. The previously identified rare variant signals for daytime sleepiness at *AR/OPHN1* (MAF = 0.002), *ROBO1* (MAF = 0.003), and *TMEM132B* (MAF = 0.004) in the first release of the UK Biobank ($N = 111,975$)[18] were not significantly associated with sleepiness in this study ($P = 0.006$–0.03; Supplementary Table 4). The lack of consistency across these analyses may relate to initial false-positive signals at rare variants (MAF = 0.001–0.005) and/or by selection bias in the initial sample in which heavy smokers had been oversampled[35]. However, two genome-wide significant loci, *HCRTR2* and *PATJ*, overlapped with those identified for a composite sleep trait in the interim release sample and a suggestive sleepiness signal at *CEPB1* was replicated. No association was seen with single-nucleotide polymorphisms (SNPs) reported in smaller independent GWAS of EDS, hypersomnia, or narcolepsy (Supplementary Table 4).

Previous longitudinal research indicated obesity and weight gain are associated with incidence of daytime sleepiness[5]; therefore, we performed an additional GWAS adjusting for BMI to identify loci that may operate in obesity-independent pathways. This analysis identified five additional loci (Supplementary Figs. 4 and 5; Supplementary Table 3). Effect estimates at the 37 loci identified in the primary model were largely unchanged.

Sensitivity analyses on autosomes additionally adjusted for potential confounders (including depression, socio-economic status, alcohol intake frequency, smoking status, caffeine intake, employment status, marital status, neurodegenerative disorders, and psychiatric problems) and stratified by obesity and sleep duration did not substantially alter effect estimates of the identified signals (Supplementary Data 1; Supplementary Tables 5 and 6). Secondary GWAS ($N = 255,426$), excluding shiftworkers and individuals with chronic health or psychiatric illnesses, additionally identified significant variants in *SEMA3D* and revealed marginally significant interactions with health status at *PATJ*, *ZENF326/BARHL2*, *ECE2*, *ASAP1*, and *CYP1A1/CYP1A2* (interaction $P < 0.05$; Supplementary Fig. 6, Supplementary Table 7). Conditional analyses at each locus identified no secondary signals. Sex-stratified analyses on autosomes additionally identified *CWC27* and *DIAPH3* in women but not in men; however, significant gene by sex interactions were not observed (Supplementary Fig. 7, Supplementary Table 8).

Replication was attempted using self-reported daytime sleepiness indices (based on related questions) available in European whites from HUNT[36] ($N = 29,906$; Supplementary Table 9) and Health 2000 studies[37] ($N = 4546$; Supplementary Table 10) (Methods). Five individual signals, including *KSR2*, *SUSD4*, and *CYP1A1/CYP1A2* were marginally significant ($P < 0.05$) and with consistent association direction in individual cohorts and/or meta-analysis (Supplementary Table 11). A genetic risk score (GRS) of 42 sleepiness loci weighted by the effect estimates from our primary daytime sleepiness GWAS was replicated in a meta-analysis of HUNT, and Health 2000 (Fisher's $P = 0.00031$; Supplementary Table 11), and this remained significant after removing the three marginally associated loci from the meta-

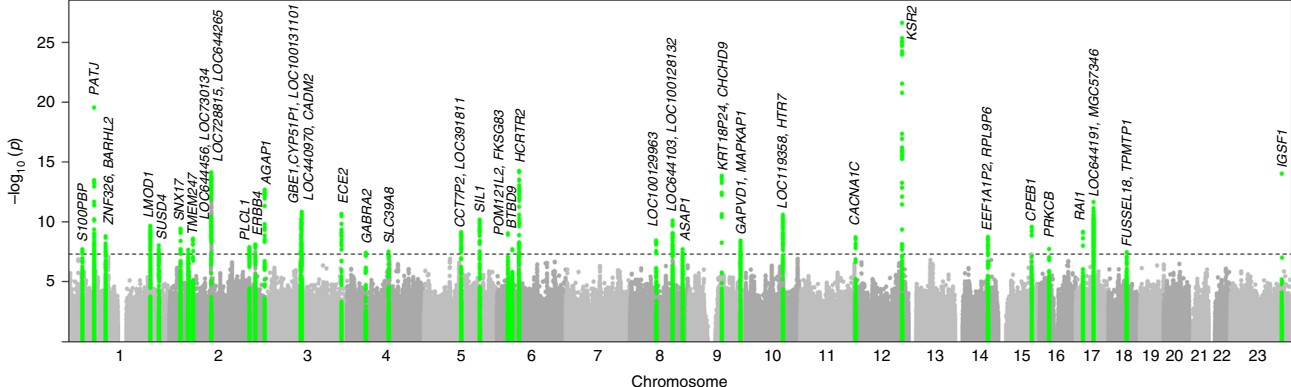

**Fig. 1** Manhattan plot for genome-wide association analysis of self-reported daytime sleepiness. Dotted line indicates genome-wide significance. Genetic association signals are highlighted in green and annotated with the nearest genes

analysis (Fisher's $P = 0.017$). Marginal association with a tiredness phenotype was observed for two signals in a meta-analysis of FINRISK[38] ($N = 20,344$; Supplementary Table 12) and Finnish Twin[39] ($N = 5766$; Supplementary Table 13); however, a combined GRS was not significant (Fisher's $P = 0.551$; Supplementary Table 14).

We further validated our results through associations with subjective and objective measures of sleep patterns and disorders available in the UK Biobank. A GRS of 42 variants was associated with self-reported shorter sleep duration, morning chronotype, increased insomnia symptoms, increased frequency of daytime napping, and with accelerometry-assessed lower sleep efficiency and increased duration of daytime inactivity (Table 1).

**Clustering of sleepiness loci suggest biological subtypes**. Genetic variants may influence daytime sleepiness through different mechanisms. Therefore, to dissect this heterogeneity, we investigated associations of individual SNPs with other sleep traits (Supplementary Data 2).

Individual daytime sleepiness increasing alleles at *PATJ* and *PLCL1* were also associated with morning chronotype; loci at metabolism regulatory genes *KSR2*, *LOC644191/CRHR1*, and *SLC39A8* with self-reported sleep duration (*KSR2* with increased sleep duration, *LOC644191/CRHR1* with long sleep, and *SLC39A8* with short sleep); *LMOD1* and *LOC644456/LOC730134* with both insomnia and short sleep duration; and at the orexin/hypocretin receptor *HCRTR2* with both morning chronotype and short sleep duration, suggesting common genetic factors. Adjusting for sleep disturbance traits (ICD-10 code defined sleep apnea or narcolepsy, or self-reported sleep duration hours, frequent insomnia symptoms or chronotype) together attenuated effect estimates for several loci, suggesting that these genetic variants influence sleepiness through altered sleep patterns and sleep disorders; however, adjustment for any trait alone only minimally altered effect estimates at individual loci (Supplementary Data 1).

Using 7-day accelerometry-derived data available in a subset of the UK Biobank ($N = 85,388$), we observed associations of several daytime sleepiness alleles with reduced sleep efficiency (e.g., *SNX17*) whereas others were associated with increased sleep efficiency (e.g., *PLCL1*), suggesting that genetic mechanisms may lead to sleepiness through effects on increased sleep fragmentation (i.e., low sleep efficiency) or increased sleep propensity (i.e., high sleep efficiency), respectively (Supplementary Data 2). Therefore, we performed hierarchical clustering analyses on risk alleles for sleepiness at 42 loci according to their association effect sizes (z-scores) with objective estimates of sleep efficiency, sleep duration, and number of sleep bouts, and self-reported frequent insomnia symptoms. An iterative approach based on silhouette

coefficients was performed to remove cluster outliers (Methods; Supplementary Fig. 8). We interpreted sleepiness alleles showing patterns of association with higher sleep efficiency, longer sleep duration, fewer discrete sleep bouts and fewer insomnia symptoms as reflective of greater intrinsic sleep propensity, whereas sleepiness alleles associated with these sleep traits in a largely inverse manner were interpreted as reflective of disturbed sleep or a sleep fragmentation phenotype resulting in less restorative sleep (Fig. 2). GRS of daytime sleepiness loci stratified by the two clusters support our interpretation, with sleep propensity loci showing robust associations with early circadian traits (e.g. morning chronotype $P = 5.54 \times 10^{-4}$; Table 1).

**Functional effects of loci**. Sleepiness loci lie in genomic regions encompassing 164 genes (Supplementary Data 3), and 3 associations are in strong linkage disequilibrium with known GWAS associations for other traits, including blood cell count, high-density liprotei cholesterol, and caffeine metabolism. Genes at multiple loci have been implicated in Mendelian syndromes or in experimental studies in mouse or fly models. Eighteen loci harbor one or more genes with potential drug targets.

We performed fine mapping analyses for potential causal variants using PICS[40] and identified 33 variants within 25 sleepiness loci with a causal probability larger than 0.2 (Supplementary Data 4). The majority of likely causal variants were intronic (65%) followed by non-coding transcript variants (8%) and nonsense-mediated decay transcript variants (7%) (Supplementary Fig. 9). Functional variants included a missense variant rs12140153 within *PATJ*, a synonymous variant rs11078398 within *RAI1*, regulatory variants rs10800796 in the promoter region of *LMOD1*, and rs239323 in a CTCF-binding site in the gene *POM121L2*. Using the Oxford Brain Imaging Genetics (BIG) server[41], we further observed the pleiotropic locus at rs13135092 (*SLC39A8*, previously associated with blood lipids, height, schizophrenia, and other traits[42]) to be significantly associated with bilateral putamen and striatum volume in the UK Biobank ($P < 2.8 \times 10^{-7}$; $N = 9,707$; Supplementary Fig. 10). This could be of particular interest given the importance of these central brain centers in influencing motor and emotional behaviors, and emerging data implicating these centers in the integration of behavioral inputs that modulate arousal and sleep–wake states[43,44].

**Gene-based, pathway, and tissue enrichment analyses**. Gene-based analyses using PASCAL[45] identified 94 genes associated with self-reported daytime sleepiness (enrichment $P < 2.29 \times 10^{-6}$) (Supplementary Table 15), of which 61 overlapped with

**Table 1 Association of weighted genetic risk score (GRS) of all 42 daytime sleepiness loci, 10 sleep propensity loci, and 27 sleep fragmentation loci with (a) multiple self-reported sleep traits and (b) 7-day accelerometry-derived sleep, circadian, and activity traits in the UK Biobank**

| Trait | All 42 sleepiness loci | | 10 sleep propensity loci | | 27 sleep fragmentation loci | |
|---|---|---|---|---|---|---|
| | Beta (SE)/OR [95% CI] per GRS effect | P value | Beta (SE)/OR [95% CI] per GRS effect | P value | Beta (SE)/OR [95% CI] per GRS effect | P value |
| **UK Biobank self-reported sleep traits** | | | | | | |
| Sleep duration (hours), n = 446,118 | −0.164 (0.053) | 0.002* | 0.642 (0.106) | $1.24 \times 10^{-9}$* | −0.586 (0.067) | $3.3 \times 10^{-18}$* |
| Long sleep duration (>8 hours), n = 34,184 cases and 305,742 controls | 1.048 [1.014,1.084] | 0.006 | 1.137 [1.064,1.216] | $1.46 \times 10^{-4}$* | 0.991 [0.95,1.034] | 0.689 |
| Short sleep duration (<7 hours), n = 106,192 cases and 305,742 controls | 1.154 [1.105,1.205] | $1.44 \times 10^{-10}$* | 0.887 [0.812,0.968] | 0.007 | 1.306 [1.235,1.382] | $7.352 \times 10^{-21}$* |
| Frequent insomnia symptomsᵃ (usually), n = 75,508 cases and 64,403 controls | 1.206 [1.132,1.284] | $6.47 \times 10^{-9}$* | 0.765 [0.674,0.868] | $3.424 \times 10^{-5}$* | 1.491 [1.376,1.617] | $2.5 \times 10^{-22}$* |
| Morning chronotype (4-level continuous variable), n = 697,828 | 0.364 (0.062) | $5.47 \times 10^{-9}$* | 0.433 (0.125) | $5.54 \times 10^{-4}$* | 0.241 (0.08) | 0.003* |
| Day naps (frequency), n = 450,918 | 0.974 (0.028) | $1.85 \times 10^{-261}$* | 1.198 (0.057) | $3.23 \times 10^{-99}$* | 0.813 (0.036) | $2.91 \times 10^{-112}$* |
| **UK Biobank 7-day accelerometry, n = 85,388** | | | | | | |
| Sleep durationᵃ (minutes) | −0.150 (0.096) | 0.116 | 1.152 (0.19) | $1.34 \times 10^{-9}$* | −0.776 (0.121) | $1.37 \times 10^{-10}$* |
| Sleep duration variability (standard deviation; minutes) | 0.069 (0.068) | 0.305 | 0.134 (0.134) | 0.317 | 0.006 (0.086) | 0.85 |
| Sleep midpoint (minutes) | −0.151 (0.059) | 0.01 | −0.285 (0.117) | 0.015 | −0.041 (0.074) | 0.579 |
| Midpoint of 5-h daily period of minimum activity (L5 timing; minutes) | −0.206 (0.12) | 0.084 | −0.651 (0.237) | $6.13 \times 10^{-3}$* | 0.18 (0.151) | 0.235 |
| Midpoint of 10-h daily period of maximum activity (M10 timing; minutes) | −0.262 (0.136) | 0.054 | −0.295 (0.27) | 0.274 | −0.118 (0.172) | 0.493 |
| Sleep efficiencyᵃ (% sleep in sleep period) | −0.028 (0.008) | $6.43 \times 10^{-4}$* | 0.065 (0.017) | $7.74 \times 10^{-5}$* | −0.067 (0.011) | $1.53 \times 10^{-10}$* |
| Number of sleep boutsᵃ (n) | −0.105 (0.405) | 0.796 | −4.231 (0.804) | $1.43 \times 10^{-7}$* | 1.722 (0.512) | $7.67 \times 10^{-4}$* |
| Daytime inactivity duration (minutes) | 0.398 (0.076) | $1.43 \times 10^{-7}$* | 0.749 (0.15) | $6.24 \times 10^{-7}$* | 0.155 (0.096) | 0.106 |

*P values significant after correction for 14 traits
ᵃSleep traits used to cluster sleep propensity and sleep fragmentation biological subtypes

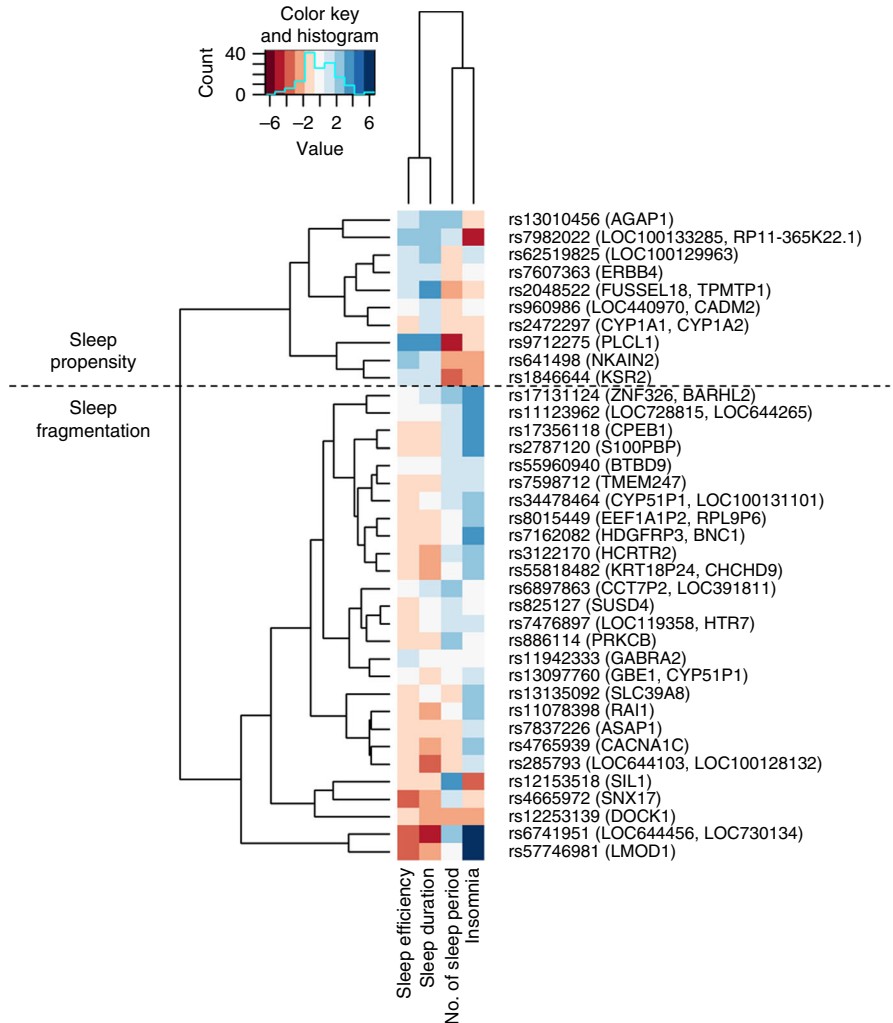

**Fig. 2** Daytime sleepiness risk alleles associate predominantly with sleep propensity or sleep fragmentation phenotypes. Each cell shows effect sizes ($z$-scores) of associations between sleepiness risk alleles (positively associated with self-reported daytime sleepiness) and sleep traits (accelerometry-derived sleep efficiency, sleep duration, number of sleep bouts, and self-reported insomnia symptoms). Blue color indicates positive $z$-scores and red color indicates negative $z$-scores. Sleep propensity alleles were defined as more likely associated with higher sleep efficiency, longer sleep duration, fewer sleep bouts, and fewer insomnia symptoms. Sleep fragmentation alleles were defined as more likely associated with lower sleep efficiency, shorter sleep duration, more sleep bouts, and more insomnia symptoms

genes under significant association peaks shown in Supplementary Data 3. Tissue enrichment analysis across 53 tissues in the GTEx database using MAGMA[46] identified multiple brain tissues including the frontal cortex, cerebellum, anterior cingulate cortex, nucleus accumbens, caudate nucleus, putamen, hypothalamus, amygdala, and hippocampus (enrichment $P < 10^{-3}$; Supplementary Table 16, Supplementary Fig. 11a). Pathway and ontology analyses using PASCAL identified enrichment in neuronal synaptic transmission pathways and EnrichR[47] identified pathways involved with the central nervous system, neurotransmitters, and metabolic processes (e.g. insulin receptor signaling pathway) (Supplementary Table 17 and Supplementary Data 5). Genes at loci showing clustering with sleep propensity phenotypes ($n = 37$) showed enriched expression in brain tissues including cortex and amygdala (enrichment $P < 10^{-3}$; Supplementary Fig. 11b). In contrast, no tissues were enriched in expression of genes that showed clustering with sleep fragmentation phenotypes ($n = 86$) perhaps reflecting further heterogeneity (Supplementary Fig. 11c). Pathway and ontology analyses results for clustered genes using FUMA also reveal different patterns (Supplementary Figs. 12 and 13).

The SNP-heritability of self-reported daytime sleepiness explained by genome-wide SNPs was estimated at 6.9% (SE = 1%). Partitioning heritability across tissue types and functional annotation classes indicated enrichment of heritability in central nervous system and adrenal/pancreas tissue lineage tissues, and in regions conserved in mammals, introns, and H3K4me1-potentially active and primed enhancers (enrichment $P < 8.3 \times 10^{-4}$) (Supplementary Tables 18 and 19).

**Genetic correlation and Mendelian randomization.** Consistent with daytime sleepiness being a symptom of several sleep disorders, GRSs of genome-wide significant SNPs for restless legs syndrome[48] ($P = 0.0002$), insomnia[49] ($P = 4 \times 10^{-7}$), and coffee consumption[50] ($P = 1.87 \times 10^{-12}$) (often used as a sleepiness "counter-measure") were significantly associated with self-reported daytime sleepiness phenotype (Table 2). Although EDS is a key symptom of narcolepsy, the GRS of narcolepsy[51] was not associated with self-reported daytime sleepiness ($P = 0.126$), suggesting narcolepsy loci did not explain sleepiness variation in this sample. We could not examine the genetic overlap of sleep apnea loci and sleepiness because few significant loci for sleep

**Table 2 Association between weighted genetic risk scores (GRS) of significant SNPs ($P < 5 \times 10^{-8}$) for other sleep behavioral traits and sleep disorders with self-reported daytime sleepiness phenotype in UK Biobank**

| Trait | N | nSNP | Beta (SE) per GRS effect | P value |
|---|---|---|---|---|
| Frequent insomnia symptoms[49] | 237,620 | 57 | 0.0007 (0.0001) | $4.00 \times 10^{-7*}$ |
| Sleep duration (hours)[59] | 446,118 | 78 | −0.0002 (0.0001) | 0.193 |
| Short sleep[59] | 106,192 | 27 | 0.0009 (0.0002) | $1.25 \times 10^{-4*}$ |
| Long sleep[59] | 34,184 | 8 | 0.0009 (0.0005) | 0.068 |
| Day naps | 450,918 | 125 | 0.4078 (0.011) | $6.61 \times 10^{-281*}$ |
| Morning chronotype[67] | 697,828 | 348 | 0.00004 (0.0001) | 0.524 |
| Restless legs syndrome[48] | 110,851 | 20 | 0.0009 (0.0002) | $2.21 \times 10^{-4*}$ |
| Narcolepsy[51] | 25,857 | 8 | 0.0007 (0.0004) | 0.126 |
| Coffee consumption (cups)[50] | 91,462 | 8 | 0.033 (0.005) | $1.87 \times 10^{-12*}$ |

*P values significant after correction for nine traits. Increasing beta reflects increasing frequency of daytime sleepiness per increase in one risk allele for other sleep trait

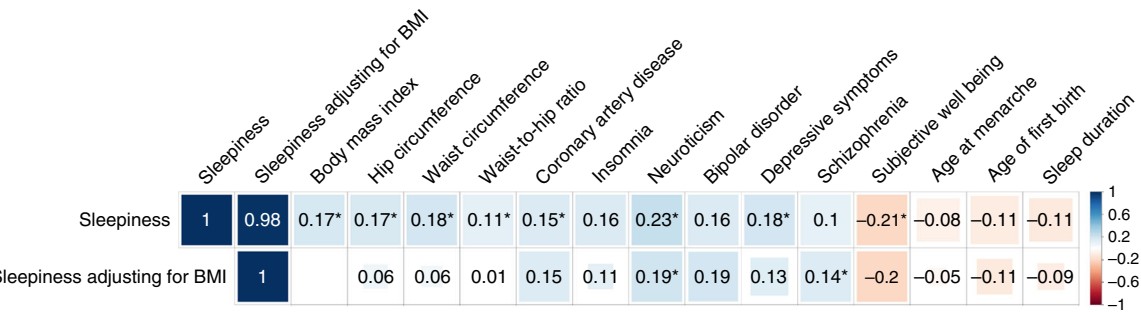

**Fig. 3** Top significant genetic correlations ($r_g$) between self-reported daytime sleepiness and published summary statistics of independent traits using genome-wide summary statistics using LD score regression (LDSC). Blue color indicates positive genetic correlation and red color indicates negative genetic correlation. Larger colored squares correspond to more significant P values, and asterisks indicate significant ($P < 2.2 \times 10^{-4}$) genetic correlations after adjusting for multiple comparisons of 224 available traits. All genetic correlations in this report can be found in tabular form in Supplementary Data 6

apnea have been reported in the literature and there was limited sleep apnea information in this cohort.

To investigate the genetic correlation between sleepiness and other common disorders, we tested the proportion of genetic variation of self-reported daytime sleepiness shared with 233 other traits with published GWAS summary statistics in LDSC[52]. After adjusting for multiple comparisons, significant positive genetic correlations were observed for daytime sleepiness with obesity traits, coronary heart disease, and psychiatric traits ($P < 0.0001$) (Supplementary Data 6). The genetic correlations with coronary artery disease and psychiatric traits persisted after adjusting for BMI (Fig. 3). Consistently suggestive negative genetic correlations for daytime sleepiness with subjective well-being and reproductive traits (age at menarche and age at first birth) were also observed ($P < 0.005$).

To evaluate the causal relationship between sleep disorders or other disease traits and daytime sleepiness, we performed two-sample summary-level Mendelian Randomization (MR) analyses using independent genetic variants from published summary statistics from GWAS of BMI, type 2 diabetes, coronary heart disease, neuroticism, bipolar disorder, depression, schizophrenia, age of menarche, restless legs syndrome, narcolepsy, insomnia, sleep duration, and chronotype as exposures and daytime sleepiness as outcome[53]. Using the inverse variance weighted (IVW) approach, we identified a putative causal association of higher BMI with increased daytime sleepiness (IVW $\beta = 0.018$; 95% CI [0.008, 0.028]; $P = 0.0004$), which was significant after accounting for multiple comparisons (IVW $P < 0.003$; Supplementary Table 20). However, there was evidence of variant heterogeneity potentially due to horizontal pleiotropy (Cochran's $Q = 677.17$;

$P = 1.09 \times 10^{-37}$; Supplementary Table 21). Therefore, we performed sensitivity analysis using the Radial MR-Egger approach (Methods)[54] to control for bias due to pleiotropy, and observed an effect that was consistent with our main IVW analyses but less precisely estimated (wider confidence intervals) because this method is statistically relatively inefficient (MR-Egger $\beta = 0.025$; 95% CI [−0.005, 0.055]; $P = 0.103$; Fig. 4 and Supplementary Table 21). An additional suggestive causal association of type 2 diabetes with increased daytime sleepiness was also observed (IVW $\beta = 0.005$; 95% CI [0.001, 0.009]; $P = 0.014$) with evidence of heterogeneity (Cochran's $Q = 88.38$; $P = 0.005$), but broadly consistent results when using Radial MR-Egger again showed a consistent effect direction (MR-Egger $\beta = 0.002$; 95% CI [−0.006, 0.01]; $P = 0.637$; Supplementary Table 21). Reverse MR did not identify any strong evidence for daytime sleepiness having a causal effect on any of the outcomes we examined (Supplementary Table 22).

## Discussion

This study expands our knowledge of the genetic architecture of daytime sleepiness. Despite the modest SNP-heritability ($h^2 = 6.9\%$, consistent with previous reports[17,18]), we identified 42 genome-wide significant loci ($P < 5 \times 10^{-8}$) given boosted statistical power with 452,071 samples. The association effects were largely unchanged adjusting for BMI, depression, socio-economic status, alcohol, smoking, caffeine, employment, neurodegenerative disorders, sleep disturbance traits individually, and upon exclusion of shiftworkers, sleep/psychiatric medication users. We did not evaluate the effect of restless legs syndrome and periodic

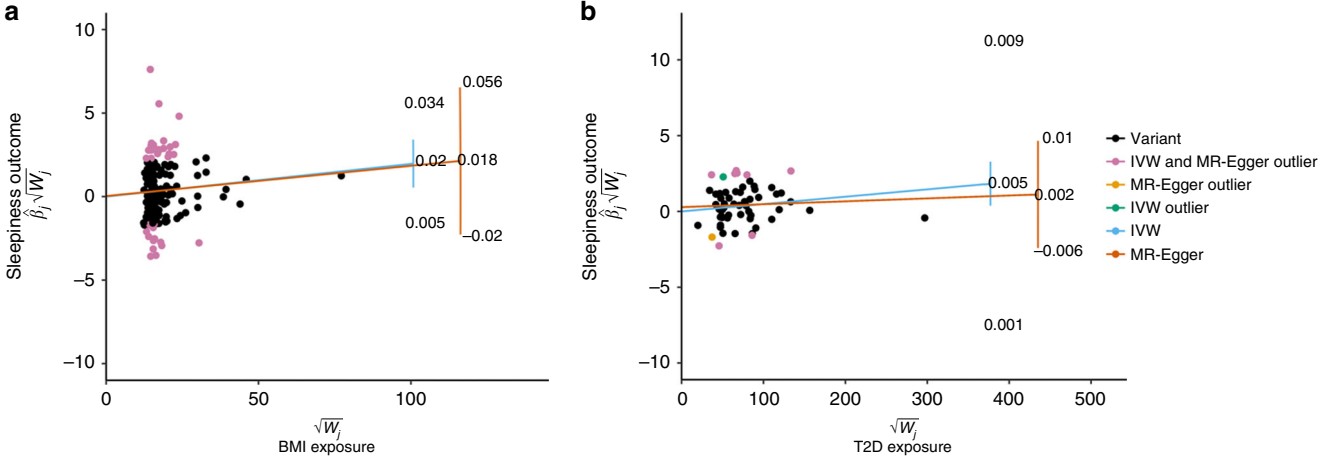

**Fig. 4** Radial plots of two-sample Mendelian randomization (MR) analysis of daytime sleepiness. **a** MR between BMI and daytime sleepiness outcome using IVW and MR-Egger tests. **b** MR between Type 2 diabetes and daytime sleepiness outcome using IVW and MR-Egger tests. The x-axis is the inverse standard error (square root weights in the IVW analysis) for each SNP. The y-axis scale represents the ratio estimate for the causal effect of an exposure on outcome for each SNP ($\hat{\beta}_j$) multiplied by the same square root weight

limb movement disorder because information on these disorders were not collected in the UKB.

An aggregate effect of a genetic risk score of 42 loci was confirmed in independent Scandinavian cohorts with different self-reported daytime sleepiness. Despite the challenges of individual loci replication with insufficient power (5–57% in replication cohorts; Supplementary Table 11), variable questionnaires in different scales across different cohorts and the multifactorial etiology of sleepiness, we observed nominal replication at five loci including our most significant association observed at KSR2, a gene regulating multiple signaling pathways (e.g. the ERK/MEK signaling pathway), affecting energy balance, cellular fatty acid, and glucose oxidation that is implicated in obesity, insulin resistance, and heart rate during sleep in previous studies in humans and mice[24,25]. While the GRS association was highly significant including the three loci with nominal significance in the meta-analysis, the effect estimates removing the three loci remained at 67% of the original effect in HUNT and 86% of the original effect in Health 2000, suggesting that additional individual sleepiness loci contribute to the combined effect of the GRS. However, replication in additional, well-powered cohorts will be important.

The validation of our results was further supported by associations between sleepiness GRS with self-reported shorter sleep duration, morning chronotype, increased insomnia risk, increased frequency of daytime napping, and with accelerometry-derived lower sleep efficiency and increased duration of daytime inactivity. These associations also suggest sleepiness loci impact other sleep parameters such as sleep latency, sleep efficiency, and sleep timing. The sleepiness GRS was not associated with 7-day accelerometry-derived continuous sleep duration, largely reflecting the heterogeneity of self-reported sleepiness (sleep propensity vs sleep fragmentation).

Our results were strengthened by previous GWAS associations with related traits (e.g. metabolic and psychiatric traits), model organism evidence for sleep phenotypes (HCRTR2, SEMA7A, and RAI1), and tissue and pathway enrichment analyses. Genes under association peaks were enriched in multiple brain tissues, including brain regions implicated in sleep–wake and arousal disorders[55] as well as centers responsive to sleep deprivation and pathways involved with the central nervous system, neurotransmitters, and metabolic processes. Enrichment of partitioning heritability were observed in variants in central nervous system

and highly conserved regions shared by human and other 28 mammals[56], suggesting strong conservation of sleep regulation throughout evolution.

We also investigated the heterogeneity of daytime sleepiness loci for the first time by performing clustering analysis according to individual SNP associations with four major sleep parameters: 7-day accelerometry-derived sleep efficiency, sleep duration, and number of sleep bouts, and self-reported frequent insomnia symptoms. We discovered risk sleepiness variants at 10 loci (e.g. PLCL1 and KSR2) and their GRS that associated with sleep propensity traits (higher sleep efficiency, longer sleep duration, fewer discrete sleep bouts, and fewer insomnia symptoms); whereas sleepiness variants at 27 loci (e.g. LMOD1, HCRTR2, and GABRA2, known to play a central role in sleep/wake control and narcolepsy[57]) and their GRS were more likely to contribute to sleep fragmentation (lower sleep efficiency, shorter sleep duration, more sleep bouts and more insomnia symptoms). Sleep propensity GRS revealed significant associations with early chronotype, reflective of circadian influences on sleep drive. Genes at sleep propensity loci also showed enriched expression in brain tissues whereas no tissues were enriched in sleep fragmentation loci, suggesting that the mechanisms associated with sleep fragmentation may be more complex, reflective of multifactorial influences. Future experimental and statistically robust clustering analysis that include other sleep and related traits are needed to validate and distinguish the biological subtypes of daytime sleepiness[58].

We extended our analysis to compare the genetic architecture between daytime sleepiness and other common disorders, and observed significant genetic correlations with obesity, coronary heart disease, and psychiatric traits. The genetic correlations of sleepiness with coronary artery disease and psychiatric traits persisted after adjusting for BMI, perhaps partially reflecting shared neurologic or neuroendocrine factors, such as those that underlay insomnia and short sleep with cardiac and psychiatric traits[49,59]. Using MR analysis, we identified potential causal association of higher BMI with increased daytime sleepiness, consistent with prospective epidemiological studies, which likely reflect metabolic and/or circadian dysfunction in obese people[5]. Suggestive causal association of type 2 diabetes and daytime sleepiness were also identified, which may reflect a high prevalence of sleep disturbances in diabetes (e.g., sleep apnea) or systemic inflammation. Reverse MR did not identify any strong

evidence for sleepiness having a causal effect on any of the outcomes we examined, implying that sleepiness is often a "symptom" of other disorders. However, replications with large samples are required to confirm the causal relationships. Future systematic MR analyses with other common disorders may be of particular interest.

This study has several strengths. It is the largest GWAS of self-reported daytime sleepiness with better power than previous studies. We identified 42 loci and confirmed the aggregated association effect in independent cohorts. Moreover, using a wide range of data on individual sleep traits—both self-reported and objectively measured—we showed that individual daytime sleepiness variants associate with unique patterns of sleep and circadian traits that largely cluster into two biological subtypes, intrinsic sleep propensity and sleep fragmentation. These findings were extended with knowledge from published databases of tissue-based expression, pathway annotations, and GWAS summary statistics for other traits.

This study also has several limitations. Primary analyses used self-reported daytime sleepiness expressed as a continuous variable derived from a 4-point scale. Future work should evaluate the psychometric properties of this question and compare it to other frequently used measures of daytime sleepiness, such as the Epworth Sleepiness Scale (ESS) or Maintenance of Wakefulness Test. It is likely that there was some loss of power due to use of a single measure of self-reported sleepiness resulting in random misclassification. However, the large sample for which questionnaire data were available provided results that were able to be further studied in a smaller sample of 7-day accelerometry-derived sleep data (which has been shown to agree well with polysomnography). Future work using objective measurements of sleepiness, such as from vigilance tests, may provide further insights into the genetics of sleepiness-related traits. The statistical power of this study may also be limited by the heterogeneity of daytime sleepiness which could be addressed by adjusting or restricting analyses using the available covariate data. Only individuals of European ancestries aged 40–69 years old in the UK were included, which limits generalizability to other populations and age groups, especially considering that sleep patterns change with age[5].

In summary, we conducted an extensive series of analysis from a large-scale GWAS and identified the heterogeneous genetic architecture of daytime sleepiness. Multiple genetic loci were identified, including genes expressed in brain areas implicated in sleep–wake control and genes influencing metabolism. Shared genetic factors were identified for daytime sleepiness and other sleep disorders, with evidence that sleepiness variants clustered with two predominant phenotypes—sleep propensity and sleep fragmentation—with the former showing stronger evidence for enrichment in central nervous system tissues, suggesting two unique mechanistic pathways. Genetic variants for daytime sleepiness also overlapped those for other diseases and lifestyle traits, with evidence that higher BMI and possibly diabetes are causally associated with increased daytime sleepiness. This work will advance understanding of biological mechanisms relating to sleepiness and underlying sleep and circadian regulation, and open new avenues for future study.

## Methods

**Population and study design.** The discovery analysis was conducted on participants of European ancestry from the UK Biobank study[22]. The UK Biobank is a prospective study that has enrolled over 500,000 people aged 40–69 living in the United Kingdom. Baseline measures collected between 2006 and 2010, including self-reported health questionnaire and anthropometric assessments were used in this analysis. Participants taking any self-reported sleep medication (Supplementary Note 1) were excluded. The UK Biobank study was approved by the National Health Service National Research Ethics Service (ref. 11/NW/0382), and all

participants provided written informed consent to participate in the UK Biobank study. In total, 452,071 individuals of European ancestry were studied with available phenotypes and genotyping passing quality control, as described below.

**EDS and covariate measurements.** Self-reported daytime sleepiness was ascertained in the UK Biobank using the question "How likely are you to doze off or fall asleep during the daytime when you don't mean to? (e.g. when working, reading or driving)" with the response options of "Never/rarely", "sometimes", "often", "all of the time", "do not know", and "prefer not to answer". Participants reporting "do not know" and "prefer not to answer" were set to missing. Other responses were coded continuously as 1 to 4 corresponding to the severity of daytime sleepiness. The primary covariates used were self-reported age and sex, and BMI calculated as weight/height[2]. Covariates used in the sensitivity analyses include potential confounders (depression, social economic status, alcohol intake frequency, smoking status, caffeine intake, employment status, marital status, neurodegenerative disorders, and use of psychiatric medications) and indices of sleep disorders and sleep traits (daytime napping, sleep apnea, narcolepsy, sleep duration, insomnia, and chronotype). Depression was recorded as a binary variable (yes/no) corresponding to question "Ever depressed for a whole week?". Social economic status was measured by the Townsend Deprivation Index based on aggregated data from national census output areas in the UK. Alcohol intake frequency was coded as a continuous variable corresponding to "daily or almost daily", "three or four times a week", "once or twice a week", "once to three times a month", "special occasions only", and "never" drinking alcohol. Smoking status was categorized as "current", "past", or "never" smoked. Caffeine intake was coded continuously corresponding to self-reported cups of tea/coffee per day. Employment status was categorized as "employed", "retired", "looking after home and/or family", "unable to work because of sickness or disability", "unemployed", "doing unpaid or voluntary work", or "full or part-time student". Neurodegenerative disorder cases ($N = 517$) were identified as a union of International Classification of Diseases (ICD)-10 coded Parkinson's disease (G20–G21), Alzheimer's disease (G30), and other degenerative diseases of nervous system (G23, G31–G32). Day napping was coded continuously ("never/ rarely", "sometimes", or "usually") responding to the question "Do you have a nap during the day?" Sleep apnea cases ($N = 5571$) were identified as a union of self-reported and ICD-10 coded (G47.3) sleep apnea. Narcolepsy cases ($N = 7$) were determined by the ICD-10 code (G47.4). Insomnia was recorded as "never/rare", "sometimes", or "usually" responding to the question "Do you have trouble falling asleep at night or do you wake up in the middle of the night?". Individuals reported "usually" were considered as frequent insomnia symptom cases. Sleep duration was recorded as discrete integers in response to the question "About how many hours sleep do you get in every 24 hours (please include naps)". In this study, short sleep was defined by sleep duration shorter than 7 h and long sleep was defined by sleep duration longer than 8 h. Chronotype was categorized as "definitely a "evening" person", "more an "evening" than a "morning" person", "more a "morning" than "evening" person", and "definitely an "morning" person". Secondary analyses were performed on participants further excluding shiftworkers, psychiatric mediation users, and participants with chronic and psychiatric illness (described in Supplementary Note 2, $N = 255,426$).

**Activity-monitor-derived measures of sleep.** Raw accelerometer data (.cwa) were collected using open source Axivity AX3 wrist-worn triaxial accelerometers (https://github.com/digitalinteraction/openmovement) in 103,711 individuals from the UK Biobank for up to 7 days[60]. We converted.cwa files to.wav files using Omconvert (https://github.com/digitalinteraction/openmovement/tree/master/Software/AX3/omconvert)[60,61]. Time windows of sleep (SPT-window) and activity levels were extracted for each 24-h period using a heuristic algorithm using the R package GGIR (https://cran.r-project.org/web/packages/GGIR/GGIR.pdf)[62,63]. Briefly, for each individual, a 5-min rolling median of the absolute change in z-angle (representing the dorsal–ventral direction when the wrist is in the anatomical position) across a 24-h period. The 10th percentile of the output was used to construct an individual's threshold, distinguishing periods with movement from non-movement. Inactivity bouts were defined as inactivity of at least 30 min duration. Inactivity bouts with less than 60 min gaps were combined to blocks. The SPT-window was defined as the longest inactivity block, with sleep onset as the start of the block and waking time as the end of the block. The sleep measurements derived from accelerometer data using this algorithm has been shown to provide reliable estimates for sleep onset time, waking time, SPT-window duration, and sleep duration within the SPT-window compared to polysomnography[63]. We applied exclusion criteria based on accelerometer data quality including (1) nonezero or missing in "data problem indicator" (Field 90002); (2) 0 in "good wear time" (Field 90015); (3) 0 in "good calibration" (Field 90016); (4) 0 in "calibrated on own data" (Field 90017); (5) "data recording errors" (Field 90182) >788 ($Q_3 + 1.5 \times IQR$); and (6) non-zero in "interrupted recording periods" (Field 90180). Accelerometry data from 85,388 participants of European ancestry passed quality control and were analyzed in this study.

The distributions of accelerometer data are described in Supplementary Table 2. The details of each measurement are as follows. L5 and M10 were the least-active 5-h window and most-active 10-h window for each day estimated from a moving average of a contiguous 5/10-h window. The L5 timing was defined as the number of hours elapsed from the previous midnight whereas M10 was defined as the

number of hours elapsed from the previous midday. Sleep midpoint was the midpoint between the start and end of the SPT-window. L5, M10, and sleep midpoint variables capture the circadian characteristics of an individual. Sleep episodes within the SPT-window were defined as periods of the $z$-axis angle change less than 5° for at least 5 min[62]. Sleep duration in an SPT-window was calculated as the sum of all sleep episodes. The mean and standard deviation of sleep duration across all SPT-windows were investigated in this study. Sleep efficiency was calculated as sleep duration divided the total SPT-window duration in an SPT-window. Sleep fragmentation was examined by counting the number of sleep episodes of at least 5 min separated by at least 5 s of wakefulness within an SPT-window. Diurnal inactivity duration was the total duration of estimated bouts of inactivity that fell outside of the SPT-window in 24 h, which included both inactivity and naps.

**Genotyping and quality control**. DNA samples of 502,631 participants in the UK Biobank were genotyped on two arrays: UK BiLEVE (807,411 markers) and UKB Axiom (825,927 markers). In all, 488,377 samples and 805,426 genotyped markers passed standard QC[64] and were available in the full data release. SNPs were imputed to a Haplotype Reference Consortium (HRC) panel (~96 million SNPs). The detailed description of genotyping, QC, and imputation are available elsewhere[64]. We further performed K-means clustering using the PCs of ~100,000 high-quality genotyped SNPs (missingness < 1.5% and MAF > 2.5%) and identified 453,964 participants of European ancestry.

**Genome-wide association analysis**. We performed a genome-wide association analysis (GWAS) of self-reported daytime sleepiness as a continuous variable derived from a 4-point scale using 452,071 individuals of European ancestry in the UK Biobank. A linear mixed regression model was applied adjusting for age, sex, genotyping array, 10 PCs, and genetic relatedness matrix, using BOLT-LMM with an MAF > 0.001, BGEN imputation score > 0.3, maximum per SNP missingness of 10%, and per sample missingness of 40%[23]. Reference 1000 genome European-ancestry (EUR) LD scores and genetic map (hg19) were implemented in this analysis. X-chromosome data were imputed and analyzed separately (with males coded as 0/2 and female genotypes coded as 0/1/2) using the same analytical approach in BOLT-LMM as was done for analysis of autosomes. A rare chrX signal at $IGSF1$ on chromosome X driven by one rare variant (MAF = 0.006) was identified, potentially attributed to genotyping artifact or false-positive association; therefore, we do not report it as a main finding. Similar linear mixed regression analyses were performed additionally adjusting for BMI and stratified by sex. Secondary GWAS excluding related individuals, shiftworkers, individuals who used psychiatric medications, and participants with chronic health and psychiatric illness ($N = 255,426$) was performed adjusting for age, sex, genotyping array, and 10 PCs in PLINK 1.9 (ref. [65]). We used a hard-call genotype threshold of 0.1, SNP imputation quality threshold of 0.80, and an MAF threshold of 0.001. SNP-heritability, defined as the proportion of trait variance explained by genome-wide additive genetic effects, was estimated using BOLT-REML[23]. Genome-wide significance level was set at $5 \times 10^{-8}$. Gene-sex and gene-health status interaction analyses were performed on unrelated individuals using a linear regression model in PLINK with the additional –interaction flag. Conditional analyses to dissect independent signals in significant genomic regions were performed using GCTA-COJO[66] with MAF > 0.001 and genome-wide significant threshold of $P < 5 \times 10^{-8}$ through a stepwise selection procedure using –cojo-slct flag. Variant annotation for each significant locus was performed using PICS with 1000 Genome EUR LD reference with a causal probability of 0.2 or greater[40].

**Sensitivity and stratification analyses of significant loci**. Sensitivity analyses of the genome-wide significant loci on autosomes in the primary analysis ($P < 5 \times 10^{-8}$) were performed additionally adjusting for potential confounders (including depression, socio-economic status, alcohol intake frequency, smoking status, caffeine intake, employment status, marital status, and psychiatric problems) and clinically important sleep traits (including sleep apnea, narcolepsy, sleep duration hours, insomnia, and chronotype) individually in 337,539 unrelated individuals using PLINK. Sleep traits were further adjusted in the model to investigate their combined effect on sleepiness signals. Stratified association analyses with self-reported daytime sleepiness in persons without obesity (BMI < 30, $N = 256,373$) vs individuals with obesity (BMI ≥ 30, $N = 81,163$), long sleepers (self-reported sleep duration > 8 h; $N = 25,272$) vs short sleepers (self-reported sleep duration < 7 h; $N = 78,393$) and tested for heterogeneity effect.

**Heterogeneity analysis**. Genome-wide significant loci identified by primary GWAS analysis were further investigated to understand their contribution to daytime sleepiness through different mechanisms by testing the associations between sleepiness risk alleles with BMI and other sleep traits in the UK Biobank (including self-reported sleep duration, insomnia, chronotype, long sleep duration [>8 h], short sleep duration [<7 h], snoring, obstructive sleep apnea defined by ICD-10 code [G47.3], hypersomnolence [defined as sleepiness plus long sleep duration without any chronic or psychiatric diseases], and 7-day accelerometry data). Linear or logistic regression analyses were performed adjusting for age, sex, genotyping array, and 10 PCs. Genome-wide summary statistics of sleep duration,

insomnia, chronotype, long sleep duration, short sleep duration, and 7-day accelerometry using BOLT-LMM were available in public database[49,59,67,68]. We performed hierarchical cluster analyses using the pairwise Euclidean distances between 42 loci: $D\left(\mathbf{X_i}, \mathbf{X_j}\right) = \sqrt{\sum_{k=1}^{4}\left(x_{ik} - x_{jk}\right)^2}$, where $\mathbf{X_i} = (x_{i1}, x_{i2}, x_{i3}, x_{i4})^{\mathrm{T}}$ corresponds to the association $z$-scores with accelerometer-derived sleep efficiency, sleep duration, sleep fragmentation (number of sleep periods), and self-reported insomnia for an SNP $i$, $i = 1, …, 42$. We took an iterative approach to improve the performance of our clustering analysis by removing cluster outliers based on silhouette coefficients. Briefly, the silhouette coefficient is a measure of how similar an object is to its own cluster (cohesion) compared to other clusters (separation). It ranges from −1 to +1, where a high value indicates that the object is well matched to its own cluster and poorly matched to neighboring clusters. In the initial clustering results (Iteration 1), loci at $GAPVD1/MAPKAP1$, $PATJ$, and $POM121L2/FKSG83$ showed negative silhouette coefficients, indicating that they were likely to be incorrectly clustered (Supplementary Fig. 8a). We therefore removed these loci in a subsequent iteration of clustering analysis. In Iteration 2, $ECE2$ locus showed a negative silhouette coefficient, and therefore this was removed (Supplementary Fig. 8b). In Iteration 3, all loci showed positive silhouette coefficients (Supplementary Fig. 8c), indicating reasonable classification. The average silhouette coefficient improved from 0.32 in our original classification to 0.4 in our final classification (with all positive coefficients).

**Gene, pathway, and tissue enrichment analyses**. We further examined the genes within genome-wide significant loci using gene-based pathway and tissue enrichment analyses[45,47,69]. Gene-based analysis was performed using PASCAL, which estimated a combined association $P$ value from the summary statistics of multiple SNPs in a gene[45]. Pathway and ontology enrichment analyses were performed using FUMA[69] and EnrichR[47]. Tissue enrichment analysis was performed using MAGMA[46] in FUMA, which controlled for gene size. Pathway and tissue enrichment analyses were also performed on genes within loci belonging to sleep propensity and sleep fragmentation clusters separately.

We constructed a weighted GRS comprising the 42 significant sleepiness loci and tested for associations with other self-reported sleep traits (sleep duration, long sleep duration, short sleep duration, insomnia, chronotype, and day naps), and 7-day accelerometry traits in the UK Biobank. Weighted GRS analyses were performed by summing the products or risk allele count multiplied by the effect estimate reported in the primary GWAS of self-reported daytime sleepiness using R package gds (https://cran.r-project.org/web/packages/gds/gds.pdf). We also tested the GRSs of reported loci for insomnia, sleep duration, short sleep, long sleep, day naps, chronotype, restless legs syndrome (RLS), narcolepsy, and coffee consumption associated with self-reported daytime sleepiness using the same approach. The SNPs selected for each trait include 57 genome-wide significant loci for frequent insomnia[49]; 78, 27, and 8 loci for sleep duration, long sleep, and short sleep, respectively[59]; 348 loci for chronotype[67]; 125 loci for daytime napping; 20 genome-wide significant loci for RLS[48]; 8 non-HLA suggestive significant loci ($P < 10^{-4}$) in a narcolepsy case–control study of European Americans[51], and 8 loci for coffee consumption[50].

**Genetic correlation analyses**. Genetic correlation analysis using LD Score regression was performed on genome-wide SNPs mapped to the HapMap3 reference panel between daytime sleepiness (with and without adjustment for BMI) and 233 published GWAS available in LDHub[52]. The significance level was determined as $10^{-4}$ correcting for multiple comparisons. Pairwise genetic correlations among daytime sleepiness, frequent insomnia, sleep duration, long sleep duration, short sleep duration, and chronotype were performed locally using LDSC. We also partitioned heritability across 8 cell-type regions and 25 functional annotation categories available in LDSC[70]. Enrichment of the partitioning heritability was calculated in each region with and without extension (±500 bp).

**MR analyses**. To investigate the causal relationship between daytime sleepiness and other traits, we performed two-sample MR using MRbase package in R[53]. IVW approach, assuming no horizontal pleiotropy effect, was implemented as the primary approach in this analysis. BMI, type 2 diabetes, coronary heart disease, psychiatric, reproductive traits, and other sleep and circadian traits (narcolepsy, insomnia, sleep duration, and chronotype) were tested as exposures for daytime sleepiness. Independent genome-wide significant SNPs extracted from publicly available summary statistics of exposures of interest (Supplementary Table 20) were tested as instruments for their effect on daytime sleepiness. The significance level was determinate as IVW $P < 0.003$ after accounting for multiple comparisons. We identified a putative causal association of higher BMI with increased sleepiness risk (IVW $P = 0.018$; 95% CI [0.008, 0.028]; $\beta = 0.0004$; Supplementary Table 21). The mean $F$ statistic was 32.7, indicating the instruments are sufficiently strong. However, Cochran's Q statistic was calculated to be 677.16 ($P = 1.09 \times 10^{-37}$), indicating substantial heterogeneity about the IVW slope. This is an indicator of potential horizontal pleiotropy that violates the traditional IV assumptions. Therefore, we applied MR-Egger regression on the radial plot scale as a sensitivity analysis[54]. The mean $I_{GX}^2$ statistic is 0.89, indicating that instruments are sufficiently strong for this analysis[71]. We observed a consistent effect direction for

Radial MR-Egger ($\beta = 0.025$; 95% CI [$-0.005, 0.055$]; $P$ value $= 0.103$). Rucker's $Q$ statistic for Radial MR-Egger is 676.58, indicating that the IVW and Radial MR-Egger models fit the data equally well (Supplementary Table 21). We also investigated the suggestive putative causal association of type 2 diabetes with increased sleepiness risk (IVW mean $F = 29$; $\beta = 0.005$; 95% CI [$0.001, 0.009$]; $P$ value $= 0.014$; Supplementary Table 21). Given variants heterogeneity evidence (Cochran's $Q = 88.38$; $P = 0.005$), we performed sensitivity analysis using Radial MR-Egger again (mean $I_{GX}^2 = 0.84$) and observed consistent effect direction ($\beta = 0.025$; 95% CI [$-0.005, 0.055$]; $P = 0.637$; Supplementary Table 21). Reverse MR between daytime sleepiness and other outcome were conducted using genome-wide significant sleepiness SNPs as instruments, and did not identify any causal association (IVW $P > 0.05$; Supplementary Table 22).

**Replication analyses**. Replication analyses were conducted using self-reported day sleepiness or fatigue in Scandinavian individuals from four population-based studies, including Nord-Trøndelag Health Study (HUNT), Health 2000 Survey, FINRISK, and The Finnish Twin Cohort Study.

The HUNT is a large longitudinal population health study, investigating the county of Nord-Trøndelag, Norway since 1984 (ref. [36]). Three surveys (HUNT1 [1984–1986], HUNT2 [1995–1997], and HUNT3 [2006–2008]) have been completed including more than 120,000 individuals. Daytime sleepiness phenotype was collected in HUNT3 by asking the question "How often in the last 3 months have you felt sleepy during the day?" with the choices "Never/seldom", "Sometimes", and "Several times". Individuals with self-reported stroke, myocardial infarction, angina pectoris, diabetes mellitus, hypo- and hyperthyroidism, fibromyalgia, and arthritis were excluded from the replication analysis.

DNA samples were collected in 71,860 HUNT samples and genotyped on one of three Illumina arrays: HumanCoreExome12 v1.0, HumanCoreExome12 v1.1, and UM HUNT Biobank v1.0. Imputation was performed on samples with European ancestries using a combined reference panel comprised of the HRC and 2202 whole-genome sequenced HUNT participants. In total, 29,906 individuals with both phenotype and imputed genotype data were available for this analysis. Sample distributions are presented in Supplementary Table 9. A generalized linear mixed model analysis was performed on continuous sleepiness adjusted for age, sex, genotyping batch effect, and four PCs using SAIGE v0.25. A second analysis additionally adjusting for BMI was conducted for replications of loci identified after adjusting for BMI.

The Health 2000 Survey is a population-based sample representing the population structure of individuals from Finland who at the time of contact were over 18 years old. Individuals over 30 years of age answered a number of health and lifestyle-related questionnaires[37]. These data were collected between 11 September 2000 and 2 March 2001 with a goal to reveal and study public health problems in Finland. The Ethics Committee of the Helsinki and Uusimaa Hospital District approved the study protocol, and a written informed consent was obtained from all participants after providing a description of the study. Full Epworth sleepiness scale (0–24) was included among the questionnaires and included from 4546 individuals with genotyping data on the study (Supplementary Table 10).

Genotyping was performed at Finnish Genome center using IlluminaHuman610K genotyping array. Imputation was performed against 2,690 hcWGS and 5092 WES Finnish genomes (http://www.sisuproject.fi/). Linear regression analyses were performed on continuous ESS adjusted for age, sex, genotyping batch effect, and 10 PCs using snptest v2.5. Shiftworkers were excluded and secondary analysis was adjusted with BMI.

The FINRISK is a population-based study initiated in 1972 and collected every 5 years since then in Finland to investigate the risk factors for cardiovascular outcomes[38]. Nine cross-sectional surveys including 101,451 participants aged 25–74 years old were conducted between 1972 to 2012. DNA samples have been collected since the 1992 survey.

We studied exhaustion and fatigue in this population. This was ascertained by asking a question "During the past 30 days, have you felt yourself exhausted or overstrained?" with choises "Never", "Sometimes" and "Often". In total, 20,344 individuals with both phenotype and whole genome genotyped and imputed data were available for this study (Supplementary Table 12). Genotyping was performed at the Wellcome Trust Sanger Institute (Cambridge, UK), at the Broad Institute of Harvard and MIT (MA, USA), and at the Institute for Molecular Medicine Finland (FIMM) Genotyping Unit using Illumina beadchips (Human610-Quad, HumanOmniExpress, HumanCoreExome). The data were imputed using the 1000 Genomes project phase 3 haplotypes and a custom haplotype set of 2000 whole genome sequenced Finnish individuals as reference panels. Linear regression analyses for exhaustion was performed with snptest v2.5 and adjusted with age, sex, genotyping batch effects, and 10 PCs. Shiftworkers were removed from analyses, A secondary analysis was additionally adjusted for BMI.

The Finnish Twin Cohort Study consists of same-sexed twin pairs born before 1958, who participated in two questionnaire surveys in 1975 and 1981. In 1990, twins who had participated in either previous survey and who were born in 1930 to 1957 were invited to participate in a questionnaire survey in 1990. The survey included a broad set of items on sleep and sleep disorders, as reported earlier[39]. Daytime fatigue was ascertained by asking the question "During the past year have you experienced any of the following symptoms: Daytime fatigue?" with the choices "Never", "Every day or almost every day", "On 3–5 days per week", "On

1–2 days per week", "Less often than once a week", "About once a month" and "Rarely". These were coded to three categories where "Never" & "Rarely" coded to represent category of "Low", "On 1–2 days per week" & "Less often than once a week" as "Intermediate" and "Every day or almost every day" & "On 3–5 days per week" as "High". Total of 5766 individuals with phenotype and imputed genotype data were available for the study (Supplementary Table 13). Genotyping were done at the Wellcome Trust Sanger Institute (Cambridge, UK), at the Broad Institute of Harvard and MIT (MA, USA), at the Institute for Molecular Medicine Finland and at the Thermo Fisher Scientific (Santa Clara CA, USA) using Illumina (Human610-Quad, Human670-QuadCustom, HumanCoreExome) and Affymetrix (FinnGen Axiom array) platforms. Genotypes were imputed using the Haplotype Reference Consortium release 1.1 reference panel. Linear mixed model association for EDS was performed with RVTESTS v2.0.9 adjusted for age, sex, and the genetic kinship matrix as a random effect controlling for sample relatedness and population structure."

A GRS of all sleepiness loci were also tested in the four cohorts. Meta-analyses of the sleepiness cohorts (HUNT and Health 2000) and the tiredness cohorts (FINRISK and Finnish Twin) were performed using Fisher's method.

**Reporting Summary**. Further information on research design is available in the Nature Research Reporting Summary linked to this article.

## Data Availability

UK Biobank Sleep Traits GWAS summary statistics are available at the Sleep Disorder Knowledge Portal (SDKP) website (http://www.sleepdisordergenetics.org). All other data are contained within the article and its supplementary information or available upon request.

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

## Acknowledgements

This research has been conducted using the UK Biobank Resource under applications 6818 and 9072. We would like to thank the participants and researchers from the UK Biobank who contributed or collected data. This work was supported by US NIH grants R01DK107859 (to R.S.), R01HL113338 (to S.R.), R35HL135818 (to S.R.), R01DK102696 (to F.S. and R.S), R01DK105072 (to R.S. and F.S.), F32DK102323 (to JML), T32HL007567(to J.M.L), K01HL135405 (to B.E.C.), R01HL127564 (to C.J.W.), R35HL135824 (to C.J.W.), and HG003054 (to X.Z.); Sleep Research Society Foundation Career Development Award 018-JP-18 (to H.W.); American Thoracic Society Foundation Unrestricted Grant (to B.E.C.); Phyllis and Jerome Lyle Rappaport MGH Research Scholar Award (to R.S.); UK Medical Research Council MC_UU_00011/3, MC_UU_00011/6, and MR/M005070/1 (to J.B., D.A.L., and M.N.W.); UK National Institute of Health Research NF-SI-0611-10196 (to D.A.L.); Diabetes UK grant 17/000570 (to M.K.R., D.A.L., and M.N.W.); the Wellcome Trust Investigator Award 107849/Z/15/Z (to D.R.); Academy of Finland #309643 (to H.M.O.); Stiftelsen Kristian Gerhard Jebsen grant (to K.H.); Research Council of Norway #231187/F20 (to B.W.); the Liaison Committee for education, research and innovation in Central Norway grant (to B.B.); the Joint Research Committee between St. Olavs hospital and the Faculty of Medicine and Health Sciences, NTNU (to K.H.); the Danish Heart Foundation 16-R107-A6779 (to J.B.N.), and the Lundbeck Foundation R220-2016-1434 (to J.B.N.). Academy of Finland grant 290039 (to T.P.). J.K. has been supported by the Academy of Finland (grants 265240, 263278, 308248, 312073).

## Author contributions

The study was designed by H.W., J.M.L., S.E.J., H.S.D., H.M.O., A.R.W., V.T.V.H., D.R., D.A.W., M.K.R., M.N.W., S.R., and R.S. H.W., J.M.L., S.E.J., H.S.D., H.O., A.R.W., V.V. H., B.B., B.W., K.Kantojärvi, T.P., Y.S., K.P., J.B., M.A.L., F.A.J.L.S., S.M.P., J.A.K, K.Kristiansson, J.B.N., D.A.L., M.K.R., M.N.W., S.R. and R.S. participated in acquisition, analysis and/or interpretation of data. H.W., J.M.L., H.S.D., H.O., K.Kantojärvi, T.P., S.R. and R.S. wrote the manuscript and all co-authors reviewed and edited the manuscript, before approving its submission. R.S. is the guarantor of this work and, as such, had full access to all the data in the study and takes responsibility for the integrity of the data and the accuracy of the data analysis.

## Additional information

**Competing interests:** H.O. is a consultant for Jazz Pharmaceuticals, Medix Biochemica, and Roche Holding. K. Kristiansson is a Consultant for Negen Ltd. F.A.J.L.S. has revieved speaker fees from Bayer Healthcare, Sentara Healthcare, Philips, Kellogg Company, Vanda Pharmaceuticals, and Pfizer. M.K.R. has acted as a consultant for Novo Nordisk and Roche Diabetes Care, and also participated in advisory board meetings on their behalf. The remaining authors declare no competing interests.

Heming Wang [1,2], Jacqueline M. Lane [2,3,4], Samuel E. Jones [5], Hassan S. Dashti [2,3,4], Hanna M. Ollila [2,3,6,7], Andrew R. Wood [5], Vincent T. van Hees [8], Ben Brumpton [9,10,11], Bendik S. Winsvold [9,12], Katri Kantojärvi [13,14], Teemu Palviainen [15], Brian E. Cade [1,2], Tamar Sofer [1,2], Yanwei Song [3,16], Krunal Patel [3,16], Simon G. Anderson [17,18], David A. Bechtold [19], Jack Bowden [10,20], Richard Emsley [19], Simon D. Kyle [21], Max A. Little [22,23], Andrew S. Loudon [19], Frank A.J.L. Scheer [1,2], Shaun M. Purcell [1,2], Rebecca C. Richmond [10,24], Kai Spiegelhalder [25], Jessica Tyrrell [5], Xiaofeng Zhu [26], Christer Hublin [15,27], Jaakko A. Kaprio [15,27], Kati Kristiansson [13], Sonja Sulkava [13,14], Tiina Paunio [13,14], Kristian Hveem [9,10], Jonas B. Nielsen [28], Cristen J. Willer [28], John-Anker Zwart [12], Linn B. Strand [9], Timothy M. Frayling [5], David Ray [29], Deborah A. Lawlor [10,20,32], Martin K. Rutter [19,30,32], Michael N. Weedon [5,32], Susan Redline [1,2,31,32] & Richa Saxena [1,2,3,4,32]

[1]Division of Sleep and Circadian Disorders, Brigham and Women's Hospital and Harvard Medical School, Boston, MA, USA. [2]Program in Medical and Population Genetics, Broad Institute, Cambridge, MA, USA. [3]Center for Genomic Medicine, Massachusetts General Hospital, Boston, MA, USA. [4]Department of Anesthesia, Critical Care and Pain Medicine, Massachusetts General Hospital and Harvard Medical School, Boston, MA, USA. [5]Genetics of Complex Traits, University of Exeter Medical School, Exeter, United Kingdom. [6]Department of Psychiatry and Behavioral Sciences, Stanford University, Palo Alto, CA, USA. [7]Institute for Molecular Medicine Finland, University of Helsinki, Helsinki, Finland. [8]Netherlands eScience Center, Amsterdam, Netherlands. [9]K.G. Jebsen Centre for Genetic Epidemiology, Department of Public Health and Nursing, Norwegian University of Science and Technology, Trondheim, Norway. [10]MRC Integrative Epidemiology Unit at the University of Bristol, Bristol, UK. [11]Department of Thoracic and Occupational Medicine, St. Olavs Hospital, Trondheim University Hospital, Trondheim, Norway. [12]Division of Clinical Neuroscience, Oslo University Hospital and University of Oslo, Oslo, Norway. [13]Genomics and Biomarkers Unit, National Institute for Health and Welfare, Helsinki, Finland. [14]Department of Psychiatry and SleepWell Research Program, Faculty of Medicine, University of Helsinki and Helsinki University Central Hospital, Helsinki, Finland. [15]Institute for Molecular Medicine FIMM, HiLIFE, University of Helsinki, Helsinki, Finland. [16]Northeastern University College of Science, Boston, MA, USA. [17]Division of Cardiovascular Sciences, School of Medical Sciences, Faculty of Biology, Medicine and Health, The University of Manchester, Manchester, UK. [18]The George Alleyne Chronic Disease Research Centre, Caribbean Institute for Health Research, University of the West Indies, Cave Hill, Barbados. [19]Division of Endocrinology, Diabetes & Gastroenterology, School of Medical Sciences, Faculty of Biology, Medicine and Health, University of Manchester, Manchester, UK. [20]Population Health Sciences, Bristol Medical School, University of Bristol, Bristol, UK. [21]Sleep and Circadian Neuroscience Institute, Nuffield Department of Clinical Neurosciences, University of Oxford, Oxford, UK. [22]Department of Mathematics, Aston University, Birmingham, UK. [23]Media Lab, Massachusetts Institute of Technology, Cambridge, MA, USA. [24]School of Social and Community Medicine, University of Bristol, Bristol, UK. [25]Clinic for Psychiatry and Psychotherapy, Medical Centre, University of Freiburg, Freiburg, Germany. [26]Department of Population and Quantitative Health Sciences, Case Western Reserve University, Cleveland, OH, USA. [27]Department of Public Health, University of Helsinki, Helsinki, Finland. [28]Department of Internal Medicine, Division of Cardiology, University of Michigan, Ann Arbor, MI, USA. [29]NIHR Oxford Biomedical Research Centre, John Radcliffe Hospital, Oxford OX39DU, UK. [30]Manchester Diabetes Centre, Manchester University NHS Foundation Trust, Manchester Academic Health Science Centre, Manchester, UK. [31]Department of Sleep Medicine, Beth Israel Deaconess Medical Center, Boston, MA, USA. [32]These authors jointly supervised this work: Deborah A. Lawlor, Martin K. Rutter, Michael N. Weedon, Susan Redline, Richa Saxena.

