## [Peer Review File · Nature Communications]

Reviewers' Comments:

Reviewer #1:

Remarks to the Author:

Review Wang et al. GWAS of EDS identified 42 loci that suggest phenotypic subgroups
Wang et al. perform a genome-wide association study (GWAS) for self-reported excessive daytime sleepiness (EDS) using the full UK Biobank dataset. Here, 452,071 participants of European ancestry provided self-reported information on the occurrence of EDS. In their main GWAS analysis of EDS frequency as a continuous variable, Wang et al. report a total of 42 risk loci. They report extensive secondary analyses including sensitivity analysis for confounders, stratified analyses by obesity, sleep duration, and sex, and a GWAS excluding shift workers and individuals with specific illnesses. The authors attempted replication of the signals using three studies with a total sample size of 54,796 individuals, but only 8 loci showed marginal significance in individual studies or a meta-analysis.

Extensive post-GWAS analyses included fine-mapping of causal variants, gene-based association tests, pathway and tissue enrichment analysis, and genetic correlation analysis. In addition, the authors performed Mendelian randomisation analyses to look at the relationship between EDS and a set of sleep disorders and other traits. For the main post-GWAS analysis, also referred to in the title, the authors used sleep measures derived from accelerometer data available in UKB. Their analysis suggests that EDS risk loci can be grouped into loci associated with greater sleep propensity and loci associated with increased sleep fragmentation, suggesting two different mechanisms leading to EDS.

In their study, Wang et al. report a significant increase in risk loci for EDS from and nominate certain brain regions and pathways implicated in neurotransmission and metabolic processes for a role in EDS. Excessive daytime sleepiness can be seen as a marker of insufficient sleep and of sleep disorders. Impaired sleep contributes substantially to public health burden. Improvements in treatment and prevention are needed. Therefore, studies such as the one by Wang et al. that offer a better understanding of the mechanisms underlying sleep are generally important.

However, I have several concerns that the authors may want to address.

Major concerns:

1). This study is an extension of a previous GWAS, which used the smaller first release of the UKB data. On page 6, lines 124/125, the authors state that three previously identified loci for EDS from this first study were confirmed now. They refer to PATJ, HCRTR2, and CPEB1. However, only CPEB1 is listed as an EDS locus in this first study (also in Supplementary Table 4 of Wang et al.), PATJ and HCRTR2 were identified for a composite phenotype "multiple sleep traits". In addition, the genome-wide significant loci from this first study are not confirmed in the current study by Wang et al. Why is this the case? One candidate locus from the previous study, DPYSL3, is not even listed in Suppl. Table 4. The inconsistency between the studies should be discussed. Is it possible that the use of a different analysis tool (BOLT-LMM) affected the results? Does coding the EDS phenotype as a continuous variable 1-4 fit well with the LMM used for quantitative traits in BOLT-LMM?

2) The GWAS does not include any data on the X chromosome. By now, this is a standard analysis in a GWAS. It is especially peculiar since one of the three genome-wide significant hits from the previous study, AR/OPHN1, is located on the X chromosome. The data are available for UKB, therefore the authors should include the X chromosome in their GWAS.

3) Replication is still of vital importance for GWAS. I find this issue especially concerning for this study as most of the loci identified as GWAS hits or candidate hits with suggestive evidence for association are not among the 42 loci reported by the authors. An attempt at replication is made, but the three datasets are considerably smaller than the discovery set and diverse phenotyping methods are used. Only eight loci replicate and not a single one across all four studies. The authors report results for a GRS built on the 42 EDS loci, suggesting that this GRS adds credibility for all identified EDS loci. But how does the GRS score perform if the replicated loci are excluded? This type of analysis is absolutely necessary to evaluate the evidence for replication. In addition, a power analysis of the individual replication samples and the meta-analysis should be given.

Supplementary table 10 only lists power estimates for HUNT study. The heterogeneity of replication for the different loci across the different studies should be discussed more clearly.

4) For me, the most interesting aspect of the manuscript is the heterogeneity analysis of the EDS loci. However, the methodology used for the clustering analysis reported in Table 1 and Figure 2, is not described in great detail. Therefore, it is difficult to judge the validity of these results. In addition, the analysis is based on using sleep measures derived from accelerometry data. Here, the data extraction and conversion is well described in the methods section. However, there are no references or analyses given which provide evidence that the derived sleep measures are accurate, e.g. by comparing them to PSG data. The authors should provide additional information on these methods.

To further support of the interesting hypothesis on two different clusters of loci – sleep propensity and sleep fragmentation - additional heterogeneity analyses within the UKB EDS sample could be of interest, e.g. using BUHMBOX to test for subgroups that could be linked to certain diseases or traits.

5) Please add references for inter-individual variability in levels of EDS (page 5, lines 92-94) and for heritability estimates in twins and families (page 5, lines 95/96).

6) Results, page 6, lines 111-117: Please indicate the strength of the observed correlations (low, moderate, strong) in the main text.

7) Abstract, page 4, line 79-80: The wording of the sentence "...indicated that higher BMI is causally associated" seems a bit too positive considering the data presented in the results section. I think this should be toned down or removed from the abstract. I really appreciate the diligent execution and detailed presentation of the MR analyses as they are presented in the results section.

8) While reading the paper, it didn't become clear to me, what was the purpose of additional analyses stratified by BMI and sleep duration? Both were also tested as a confounder according to Supplementary Table 5. I found it quite hard sometimes to follow the main story of the manuscript due to the large number of analyses presented. Maybe some could be moved to supplemental material if they only support conclusions which can already be drawn based on one of the other analyses presented.

9) A limitation of the study is that there is little to no detailed information on the impact of two important and also frequent sleep-related disorders which can lead to EDS, restless legs syndrome and periodic limb movement disorder, in the correlation and sensitivity analyses. The authors could acknowledge that their selection of traits is limited by the availability of data in UKB.

10) Neurodegenerative disorders frequently affect sleep and lead to EDS. Were these included in the sensitivity analyses? If not, they should be included in the confounder analysis. If data in UKB is available, it would be interesting to see if having children acts as a confounder or not.

11) Supplementary Table 12 presents annotation information. It is called gene annotation. However, a description how this table was compiled is missing. It is not clear, how loci were defined, how the genes listed in the table were selected, or if the GWAS catalogue was screened only for lead SNPs or also for their LD-proxies. The latter approach could result in an increased overlap with the GWAS catalogue for the EDS SNPs. The software (?) PheGeni is mentioned, but no description/reference is presented. However, such information is of importance, especially for readers not that familiar with GWAS methodology.

Minor concerns:

1) Title: I am not sure if "phenotypic subgroup" is the best term to use. From my point of view, phenotypic would refer to the self-reported EDS phenotype itself. Therefore, I would expect data that identified different phenotypic presentations of EDS in your UKB sample. However, your analyses indicate different mechanisms leading to an EDS phenotype. Therefore, I would suggest to rephrase.

2) Method description is concise, but details on important individual specific settings used in programs such as BOLT-LMM, PLINK, GCTA-COJO, PICS etc. should be reported in the supplement. Readers would benefit, because it is easier to understand the analyses. Scientists working on similar phenotypes would benefit as well, because they could analyse their own datasets with the same methods in order to ensure comparability of the results.

- 3) Regional association plots for the 42 risk loci could be included as supplementary figures.
- 4) Numbers reported for the accelerometry-derived data differ: 85,388 (page 9, line 179), 85,499 (table 1) and 85,502 (page 20, line 434).
- 5) Supplementary Table 2: For $P=0$, the exact values should be given.
- 6) Supplementary Table 4: The header reads "Replication in UK Biobank of genetic variants reported in prior studies to be associated with excessive daytime sleepiness and other related traits." However, some of the loci in this table were not genome-wide significant in the previous GWAS, therefore this should not be called "replication". A more neutral expression could be used such as "EDS GWAS association results of....".
- 7) Supplementary Table 5: typo, Sensitivity
- 8) Supplementary Figure 6: typo?, (optional)
- 9) To avoid confusion, heritability estimates calculated from GWAS data should be referred to as "SNP-heritability", e.g. page 11, line 231 or page 13, line 278.
- 10) typo?: restless leg syndrome should be called restless legs syndrome, according to ICD-10, DSM-5.
- 11) Supplementary Table 8 reports two genes identified in the sex-specific GWAS, but these results are not mentioned in the main text.
- 12) For the main GWAS, self-reported EDS is used as the phenotype. This is important information and should be included in the title if possible.
- 13) Page 10, line 208: What makes the locus at rs13135092 pleiotropic? It was associated with self-reported sleep duration in a previous study, and with EDS in the current study. However, what evidence is there for this being a pleiotropic effect and not EDS due to changes in sleep duration, which would reflect a shared pathway.

Reviewer #2:

Remarks to the Author:

This manuscript reports on a genome-wide association study of daytime sleepiness. Despite some intriguing findings, there are several major concerns that need to be addressed.

1. The authors used the term "excessive daytime sleepiness (EDS)" in the manuscript. EDS generally indicates chronic and pathogenic sleepiness. The following question was utilized to confirm daytime sleepiness on page 18: "How likely are you to doze off or fall asleep during the daytime when you don't mean to? (e.g. when working, reading or driving)" with the response options of "Never/rarely", "sometimes", "often", "all of the time". The question is not appropriate for ascertaining EDS. Epworth sleepiness scale (ESS), multiple sleep latency test (MSLT) and/or 24hr polysomnography (PSG) are needed to assess EDS. Therefore, I recommend that the term not "EDS" but "daytime sleepiness" should be used in this manuscript.

The title of this article should also include the term "self-reported". It is important for sleep scientists to know whether subjective or objective measures were employed to assess daytime sleepiness when they check the title.

("Genome-wide association analysis of self-reported daytime sleepiness identifies 42 loci that suggest phenotypic subgroups")

2. When sleep clinicians assess and diagnose EDS in patients, they have to consider insufficient sleep (Sleep. 2014; 37(6): 1035–1042.). Sleep scientists also need to treat insufficient sleep as an important confounding factor to appropriately obtain sleep data from subjects with pathogenic daytime sleepiness. Therefore, the author should discuss insufficient sleep in this manuscript. For example, "Did the UK biobank (UKB) collect data of insufficient sleep for the subjects?", "How did insufficient sleep affect the results of this study?", "Was it possible to control for the effect of insufficient sleep?" and so on.

3. I was honestly a little surprised at results of replication studies in three cohorts. Only 8 loci out of 42 tested loci were replicated in the three cohorts. Probably, the 8 loci showed nominally

significant associations because the Methods did not mention that a multiple testing correction such as bonferroni correction was applied for the replication studies (although 126 times were tested). In addition, effects of several SNPs, +/- of β values, were opposite between the UKB and replication studies in the Table S10. In my knowledge, genetic variants identified by self-reported chronotype GWASs were more replicated in independent studies (Nat Commun. 2016 9; 7:10889, Supplementary Table 3.).

The authors described "Despite the challenges of individual loci replication with insufficient power, variable questionnaires across different cohorts and the multi-factorial etiology of EDS" on page 14 in the manuscript. I recognize the importance of sample sizes, however I consider that differences in questions between studies would affect the association results. Some readers might think that GWASs of daytime sleepiness are easily influenced by question content and question wording. The authors should more discuss this point because this is related to the reliability of the UKB daytime sleepiness questionnaire. For example, "How can differences in questionnaires be overcome?", "Is the validation of the questionnaire using MSLT, PSG or ESS necessary in a future study" and so on.

4. I consider that rs7598712 was not replicated in the Finrisk although the p value was 0.045 in the Table S10. The reason is that the β value of the UKB was 0.006, while that of the Finrisk was -0.012. These effects seem to be opposite.

5. The authors performed a linear mixed regression model using EDS as a continuous variable of 4 integers (on Page 22). The number of subjects who answered "all of the time" was only 29. Are there possibilities that the small sample size causes problems for your analyses? Is it reasonable for regression analyses to bring two groups, "often" and "all of the time", into one category?

6. In the Table S14, more genes were identified by a gene-based association analysis, compared to 42 loci. Associated genetic variants in each significant gene were not included in the Table S14. Could the authors give us the information of the genetic variants with annotation? (Were there so many SNPs? Did pascal compute only gene scores and gene p-values?)

7. I do not understand the following sentence on page 11: "Gene-based analyses using PASCAL identified 94 genes associated with EDS ($p < 2.29 \times 10^{-6}$) (Supplementary Table 14), of which 63 overlapped with genes under significant association peaks shown in Supplementary Table 13." I could not find the 63 genes.

Response to reviewers' comments

We would like to thank the referee for their constructive and thoughtful comments, which we address below. Reviewer comments are quoted in italics and our point-by-point responses follow in blue. Modifications and additions to the manuscript are included in our responses to the reviewers and are highlighted in red in the revised manuscript.

Reviewer #1 (Remarks to the Author):

Wang et al. GWAS of EDS identified 42 loci that suggest phenotypic subgroups Wang et al. perform a genome-wide association study (GWAS) for self-reported excessive daytime sleepiness (EDS) using the full UK Biobank dataset. Here, 452,071 participants of European ancestry provided self-reported information on the occurrence of EDS. In their main GWAS analysis of EDS frequency as a continuous variable, Wang et al. report a total of 42 risk loci. They report extensive secondary analyses including sensitivity analysis for confounders, stratified analyses by obesity, sleep duration, and sex, and a GWAS excluding shift workers and individuals with specific illnesses. The authors attempted replication of the signals using three studies with a total sample size of 54,796 individuals, but only 8 loci showed marginal significance in individual studies or a meta-analysis. Extensive post-GWAS analyses included fine-mapping of causal variants, gene-based association tests, pathway and tissue enrichment analysis, and genetic correlation analysis. In addition, the authors performed Mendelian randomisation analyses to look at the relationship between EDS and a set of sleep disorders and other traits. For the main post-GWAS analysis, also referred to in the title, the authors used sleep measures derived from accelerometer data available in UKB. Their analysis suggests that EDS risk loci can be grouped into loci associated with greater sleep propensity and loci associated with increased sleep fragmentation, suggesting two different mechanisms leading to EDS. In their study, Wang et al. report a significant increase in risk loci for EDS from and nominate certain brain regions and pathways implicated in neurotransmission and metabolic processes for a role in EDS. Excessive daytime sleepiness can be seen as a marker of insufficient sleep and of sleep disorders. Impaired sleep contributes substantially to public health burden. Improvements in treatment and prevention are needed. Therefore, studies such as the one by Wang et al. that offer a better understanding of the mechanisms underlying sleep are generally important.

We thank the reviewer for this assessment.

However, I have several concerns that the authors may want to address.

Major concerns:

1). This study is an extension of a previous GWAS, which used the smaller first release of the UKB data. On page 6, lines 124/125, the authors state that three previously identified loci for EDS from this first study were confirmed now. They refer to PATJ, HCRTR2, and CPEB1. However, only CPEB1 is listed as an EDS locus in this first study (also in Supplementary Table 4 of Wang et al.), PATJ and HCRTR2 were identified for a composite phenotype “multiple sleep traits”.

We agree that this is confusing and now confirm validation of CPEB1 as the only previous EDS locus in the main text. In the smaller release, “Multiple trait” analysis combining sleepiness, insomnia, sleep duration, and chronotype identified 6 loci, of which the associations of PATJ and HCRTR2 were primarily

driven by sleepiness ($P < 10^{-5}$), while the associations of *AK5*, *RGS16*, *MEIS1*, and *PAX8* were driven by other sleep traits ($P > 0.1$ with sleepiness; Table R1). We have now removed these multi-trait loci from Supplementary Table 4 to avoid confusion.

In addition, the genome-wide significant loci from this first study are not confirmed in the current study by Wang et al. Why is this the case? One candidate locus from the previous study, DPYSL3, is not even listed in Suppl. Table 4. The inconsistency between the studies should be discussed.

Is it possible that the use of a different analysis tool (BOLT-LMM) affected the results? Does coding the EDS phenotype as a continuous variable 1-4 fit well with the LMM used for quantitative traits in BOLT-LMM?

We appreciate the reviewer's important suggestion to discuss the inconsistency between current (full UKBB) and previous (interim release) UKB GWAS.

First, we clarify that the prior interim release analysis, "single trait" analysis for sleepiness identified only one genome-wide significant locus in primary analysis (*AR/OPHN1*), one in a secondary analysis adjusted for depression (*ROBO1*), and a third in a secondary analysis adjusted for BMI (*TMEM132B*; $P < 5 \times 10^{-8}$). We added chromosome X association analysis as suggested by the reviewer and observed only a weak association with rs7356079 at *AR/OPHN1* ($P = 5.9 \times 10^{-3}$). Other loci only met suggestive criteria for significance; thus, it is not unexpected that some might be false positive that would not be replicated. Among the suggestive loci, rs35309287 at *DPYSL3* was not tested because of low imputation quality in the full UKBB dataset (info score=0.023). We looked up a proxy variant (rs34398961; $r^2=1$ in 1000G EUR) in full UKBB analysis, which is not significant ($P=0.076$) (Table R1). We now limit discussion of replication to the three genome-wide significant loci. In Table R1 below we summarize the top SNPs in the interim analysis and in the full analysis.

There are several reasons for lack of replication:

- ***Rare variants***: The three genome-wide significant variants in the interim analysis were very rare (MAF=0.001-0.005), increasing their likelihood to be false positives. These two sets of analyses underscore the need to interpret results of very low frequency variants with caution until replicated in independent data, confirmed by sequencing or from animal or cellular model studies.
- ***Ascertainment bias***: We categorized sleepiness using the same coding as in phase I analysis (1-4 corresponding to "never/rarely", "sometimes", "often", "all the time"). The distributions of sleepiness between interim release and full UKBB are in general similar. However, the interim release sample was selected for studying smoking, lung function, and COPD, which includes ~25,000 heavy smokers with mean 35 pack-year (Wain, Shrine et al. 2015). The ascertainment of the interim release data may have introduced selection bias.
- ***Analysis consistency***: In the full UKBB analysis, we performed BOLT-LMM on related samples (Supplementary Table 3) and PLINK on unrelated samples as secondary analyses (Supplementary Table 5). The results were not substantially different, indicating consistency between the analysis tools. BOLT-LMM did not assume a specific trait distribution, but the results may be skewed for a quantitative trait with extreme outliers or an unbalanced case-control design with less than 10% case fraction (Loh, Kichaev et al. 2018), which is not the case in this analysis.

We now added in Results “The previously identified rare variant signals for daytime sleepiness at *AR/OPHN1* (MAF=0.002), *ROBO1* (MAF=0.003) and *TMEM132B* (MAF=0.004) in the first release of the UK Biobank (N=111,975) (Lane, Liang et al. 2017) were not significantly associated with sleepiness in this study (P=0.006-0.03; Supplementary Table 4). The lack of consistency across these analyses may relate to initial false positive signals at rare variants (MAF=0.001-0.005) and/or by selection bias in the initial sample in which heavy smokers had been oversampled(Wain, Shrine et al. 2015). However, two genome-wide significant loci, *HCRTR2* and *PATJ*, overlapped with those identified for a composite sleep trait in the interim release sample.” (Page 7)

Table R1: Side by side comparison of associations between interim release analysis and full UKBB analysis.

SNP	Nearest Gene(s)	Alleles (E/A)	Interim release analysis					Full UKBB analysis				
			EA	INFO	BETA	SE	P	EA	INFO	BETA	SE	P
rs73536079	AR, OPHN1	C/G	0.998	0.9	-0.634	0.115	3.94×10 ⁻⁸	0.999	0.96	-0.092	0.033	5.9×10 ⁻³
rs182765975	ROBO1	G/T	0.997	0.86	-0.099	0.018	3.33×10 ⁻⁸	0.996	0.87	-0.023	0.008	0.0073
rs142261172*	TMEM132B	G/A	0.996	0.92	-0.106	0.018	9.06×10 ⁻⁹	0.995	0.93	-0.016	0.008	0.042
rs192315283	HSD52	T/C	0.99	0.76	-0.126	0.025	3.55×10 ⁻⁷	0.991	0.83	-0.016	0.006	0.008
rs76645968	ASB3	G/C	0.977	0.99	0.073	0.014	1.79×10 ⁻⁷	0.976	0.99	0.013	0.003	2×10 ⁻⁴
rs920065	MRPS35P1, MRPS36P1	G/C	0.176	0.96	-0.028	0.006	4.25×10 ⁻⁷	0.176	0.96	-0.003	0.001	0.028
rs115320831	TMEM144	G/A	0.702	0.98	0.024	0.005	3.68×10 ⁻⁷	0.729	0.95	0.001	0.001	0.27
rs34398961	DPYSL3	C/A	0.989	0.94	0.125	0.026	1.1×10 ⁻⁶	0.992	0.62	0.012	0.007	0.076
rs189689339	FAM46A	C/T	0.997	0.67	-0.226	0.044	2.13×10 ⁻⁷	0.997	0.87	-0.020	0.010	0.047
rs17507216	CPEB1	G/A	0.768	1	-0.026	0.005	1.59×10 ⁻⁷	0.768	1	-0.008	0.001	2.90×10 ⁻¹⁰
rs12140153	PATJ	T/G	0.901	0.93	0.036	0.007	6.60×10 ⁻⁷	0.905	0.95	0.017	0.002	2.80×10 ⁻²⁰
rs3122163	HCRTR2	T/C	0.232	0.99	0.023	0.005	5.51×10 ⁻⁶	0.231	0.99	0.010	0.001	5.70×10 ⁻¹⁵
rs76681500	AK5	A/G	0.841	0.99	0.008	0.006	0.15	0.840	1.00	0.001	0.001	0.67
rs694383	RGS16	C/G	0.030	1.00	-0.009	0.012	0.47	0.031	1.00	-0.006	0.003	0.044
rs113851554	MEIS1	T/G	0.944	1.00	0.002	0.009	0.85	0.943	0.93	-0.001	0.002	0.55
rs62158211	PAX8	T/G	0.786	0.99	-0.005	0.005	0.37	0.787	0.99	0.000	0.001	0.94

*The association with rs142261172 in interim release and full UKBB analysis adjusted for BMI.

2) The GWAS does not include any data on the X chromosome. By now, this is a standard analysis in a GWAS. It is especially peculiar since one of the three genome-wide significant hits from the previous study, *AR/OPHN1*, is located on the X chromosome. The data are available for UKB, therefore the authors should include the X chromosome in their GWAS.

We thank the reviewer for this suggestion. We now added the association analysis for the X chromosome and identified rs189568347 at *IGSF1* to be significantly associated with sleepiness (beta=-0.046; P=9.4e-15). We have updated Supplementary Table 3 and Figures. Interestingly, this signal is just driven by one rare variant (MAF=0.006), therefore we cannot rule out genotyping artifact or a false positive (see comments above). We now added in the Methods “A rare chrX signal at *IGSF1* on chromosome X driven by one rare variant (MAF=0.006) was identified, potentially attributed to genotyping artifact or false positive association, therefore we do not report it as a main finding.”. (Page 24). We did not pursue subsequent characterization including sensitivity, heterogeneity, and replication analyses on this locus because of our limited confidence in this finding.

3) Replication is still of vital importance for GWAS. I find this issue especially concerning for this study as most of the loci identified as GWAS hits or candidate hits with suggestive evidence for association are not among the 42 loci reported by the authors. An attempt at replication is made, but the three datasets are considerably smaller than the discovery set and diverse phenotyping methods are used. Only eight loci replicate and not a single one across all four studies. The authors report results for a GRS built on the 42 EDS loci, suggesting that this GRS adds credibility for all identified EDS loci. But how does the GRS score perform if the replicated loci are excluded? This type of analysis is absolutely necessary to evaluate the evidence for replication. In addition, a power analysis of the individual replication samples and the meta-analysis should be given. Supplementary table 10 only lists power estimates for HUNT study. The heterogeneity of replication for the different loci across the different studies should be discussed more clearly.

While we fully appreciate the concern about replication, we note that the discovery dataset is magnitudes larger than prior studies of sleepiness or available independent cohorts. Notably, the power to replicate individual SNPs in the replication cohorts is very low - the HUNT study has $\sim 1/15^{\text{th}}$ the sample size of UKBB (power 18-57%) and the Health 2000 has $\sim 1/100^{\text{th}}$ (power $\sim 5\%$ for all variants). We have now added the power calculation for other replication cohorts, and separated the results for replication in genetic analysis of sleepiness questions (HUNT and Health 2000) from a related tiredness phenotype (FINRISK and Finnish Twin). We now reported "Five individual signals, including KSR2, SUSD4, and CYP1A1/CYP1A2 were marginally significant ($p < 0.05$) and with consistent association direction in individual cohorts and/or meta-analysis". (Page 8)

Based on the Reviewer's suggestion, we tested the GRS after excluding the 3 marginally associated loci from the meta-analysis. The association remained significant in the meta-analysis of HUNT and Health 2000 ($P = 0.017$). (Page 8)

4) For me, the most interesting aspect of the manuscript is the heterogeneity analysis of the EDS loci. However, the methodology used for the clustering analysis reported in Table 1 and Figure 2, is not described in great detail. Therefore, it is difficult to judge the validity of these results. In addition, the analysis is based on using sleep measures derived from accelerometry data. Here, the data extraction and conversion is well described in the methods section. However, there are no references or analyses given which provide evidence that the derived sleep measures are accurate, e.g. by comparing them to PSG data. The authors should provide additional information on these methods.

We thank the reviewer for these helpful comments. We added more details for clustering analysis in the methods: "We performed hierarchical cluster analyses using the pairwise Euclidean distances between 42 loci: $D(X_i, X_j) = \sqrt{\sum_{k=1}^4 (x_{ik} - x_{jk})^2}$, where $X_i = (x_{i1}, x_{i2}, x_{i3}, x_{i4})^T$ corresponds to the association z-scores with accelerometer derived sleep efficiency, sleep duration, sleep fragmentation (number of sleep periods), and self-reported insomnia for a SNP i , $i = 1, \dots, 42$." (Page 26)

We appreciate the reviewer's concern about validation of the analysis. The sleep measurements derived from accelerometer data using a heuristic algorithm have been shown to provide reliable estimates for

sleep onset time, waking time, SPT-window duration, and sleep duration within the SPT-window compared to polysomnography. (van Hees, Sabia et al. 2018) (Page 22)

We also took an iterative approach to improve the performance of our clustering analysis by removing cluster outliers based on the silhouette coefficient. Briefly, the silhouette coefficient is a measure of how similar an object is to its own cluster (cohesion) compared to other clusters (separation). It ranges from -1 to +1, where a high value indicates that the object is well matched to its own cluster and poorly matched to neighboring clusters. In our original clustering results, loci at *GAPVD1/MAPKAP1*, *PATJ(INADL)*, and *POM121L2/FKSG83* showed negative silhouette coefficients, indicating that they were likely to be incorrectly clustered (Fig. R1A). We therefore removed these loci in a subsequent iteration of clustering analysis. In Iteration 2, *ECE2* locus then showed a negative silhouette coefficient, and therefore this was removed (Fig. R1B). In Iteration 3, all loci showed positive silhouette coefficients (Fig. R1C), indicating reasonable classification. The average silhouette coefficient improved from 0.32 in our original classification to 0.4 in our final classification (with all positive coefficients).

We have added the details of this analysis in the methods (Page 26) and results (Page 10) sections. We have updated the final classification heatmap (Fig. 2) and the cluster stratified GRS, tissue and pathway enrichment analyses (Table 1, Supplementary Fig. 10-12). The results were not largely changed. Notably, additional validation using external data will be needed to confirm the robustness of the heterogeneity effects.

To further support of the interesting hypothesis on two different clusters of loci – sleep propensity and sleep fragmentation - additional heterogeneity analyses within the UKB EDS sample could be of interest, e.g. using BUHMBOX to test for subgroups that could be linked to certain diseases or traits.

We performed pathway and tissue enrichment analyses of the two classified variant groups and observed different enrichment patterns. “Genes at sleep propensity loci also showed enriched expression in brain tissues whereas no tissues were enriched in sleep fragmentation loci, suggesting that the mechanisms associated with sleep fragmentation may be more complex, reflective of multifactorial influences.” (Page 16). We agree with the reviewer that additional heterogeneity analysis that identifies subgroups linked to certain diseases and traits is interesting and important as a future research direction, however given the lack of publicly available large scale GWAS for RLS, sleep apnea and narcolepsy, this type of BUHMBOX analysis is beyond the scope of the current manuscript.

Figure R1. Clustering silhouette plots of three iterations. Loci with negative silhouette coefficients were removed in the next iteration of clustering analysis. (Supplementary Fig 7)

5) Please add references for inter-individual variability in levels of EDS (page 5, lines 92-94) and for heritability estimates in twins and families (page 5, lines 95/96).

We have now added the references for inter-individual variability in levels of sleepiness (Van Dongen, Baynard et al. 2004, Kapur, Baldwin et al. 2005) and for heritability estimated in twins (Carmelli, Bliwise et al. 2001, Desai, Cherkas et al. 2004, Van Dongen, Vitellaro et al. 2005) and families (Watson, Goldberg et al. 2006). (page 5)

6) Results, page 6, lines 111-117: Please indicate the strength of the observed correlations (low, moderate, strong) in the main text.

All of our observed correlations are low (Spearman correlation < 0.2). We now add “weakly correlated” in the main text. (Page 6)

7) Abstract, page 4, line 79-80: The wording of the sentence “...indicated that higher BMI is causally associated” seems a bit too positive considering the data presented in the results section. I think this should be toned down or removed from the abstract. I really appreciate the diligent execution and detailed presentation of the MR analyses as they are presented in the results section.

We now removed the MR results from the abstract. (Page 4)

8) While reading the paper, it didn't become clear to me, what was the purpose of additional analyses stratified by BMI and sleep duration? Both were also tested as a confounder according to Supplementary Table 5. I found it quite hard sometimes to follow the main story of the manuscript due to the large number of analyses presented. Maybe some could be moved to supplemental material if they only support conclusions which can already be drawn based on one of the other analyses presented.

We thank the reviewer for this advice. The stratification analysis is part of the sensitivity analysis. We now simplify the text as “Sensitivity analyses additionally adjusted for potential confounders (...) and stratified by obesity and sleep duration did not substantially alter effect estimates for the 42 identified signals (Supplementary Table 5 and 6). (page 8)

9) A limitation of the study is that there is little to no detailed information on the impact of two important and also frequent sleep-related disorders which can lead to EDS, restless legs syndrome and periodic limb movement disorder, in the correlation and sensitivity analyses. The authors could acknowledge that their selection of traits is limited by the availability of data in UKB.

Unfortunately, the UKB does not have measures for restless legs syndrome or periodic limb movement disorder, therefore we cannot investigate the effect in correlation and sensitivity analyses. This study did investigate the genetic risk score of previously published restless legs syndrome loci and identified associations with sleepiness (Table 2). We now added in the discussion “We did not evaluate the effect of restless legs syndrome and periodic limb movement disorder because information on these disorders were not collected in the UKB”. (Page 15)

10) Neurodegenerative disorders frequently affect sleep and lead to EDS. Were these included in the sensitivity analyses? If not, they should be included in the confounder analysis. If data in UKB is available, it would be interesting to see if having children acts as a confounder or not.

Thank you for this suggestion. We performed another sensitivity analysis adjusting for neurodegenerative disorders. “Neurodegenerative disorder cases (N=517) were identified as a union of ICD-10 coded Parkinson’s disease (G20-G21), Alzheimer’s disease (G30), and other degenerative diseases of nervous system (G23, G31-G32)” (Page 20). This analysis did not alter effect estimates of the 42 identified signals. (Page 8 and Supplementary Table 5).

11) Supplementary Table 12 presents annotation information. It is called gene annotation. However, a description how this table was compiled is missing. It is not clear, how loci were defined, how the genes listed in the table were selected, or if the GWAS catalogue was screened only for lead SNPs or also for their LD-proxies. The latter approach could result in an increased overlap with the GWAS catalogue for the EDS SNPs. The software (?) PheGeni is mentioned, but no description/reference is presented. However, such information is of importance, especially for readers not that familiar with GWAS methodology.

We changed the table title as: “Annotation of genes under association signals for self-reported daytime sleepiness.” We also added a column of “Locus order” and added a footnote “Locus order corresponds to the order of loci reported in Supplementary Table 3. The association interval of each locus was defined as the boundary for all SNPs with $r^2 \geq 0.5$ to lead SNP in 1KG CEU. The genes within that region were annotated using SNPsea (Slowikowski, Hu et al. 2014). For annotated genes, GWAS hits with $r^2 > 0.7$ were identified by the NHGRI GWAS catalog (MacArthur, Bowler et al. 2017). Gene expression enrichment in GTEX was evaluated relative to non-brain tissues and the databases PheGenI (Ramos, Hoffman et al. 2014). OMIM (Hamosh, Scott et al. 2005), DGIdb (Cotto, Wagner et al. 2018) were queried for each gene for reported Mendelian diseases and druggable targets. A list of genes with published sleep phenotypes in model organisms (mouse, flies, worms, zebrafish) were queried and queried.”

Minor concerns:

1) Title: I am not sure if “phenotypic subgroup” is the best term to use. From my point of view, phenotypic would refer to the self-reported EDS phenotype itself. Therefore, I would expect data that identified different phenotypic presentations of EDS in your UKB sample. However, your analyses indicate different mechanisms leading to an EDS phenotype. Therefore, I would suggest to rephrase.

We changed the title to “Genome-wide association analysis of self-reported daytime sleepiness identifies 42 loci that suggest heterogenous subgroups”.

2) Method description is concise, but details on important individual specific settings used in programs such as BOLT-LMM, PLINK, GCTA-COJO, PICS etc. should be reported in the supplement. Readers would benefit, because it is easier to understand the analyses. Scientists working on similar phenotypes would benefit as well, because they could analyse their own datasets with the same methods in order to ensure comparability of the results.

We added more detailed descriptions of the settings for the genetic software in Methods (Page 23-24):

BOLT-LMM: “We performed a genome-wide association analysis (GWAS) of self-reported daytime sleepiness as a continuous variable derived from a 4-point scale using 452,071 individuals of European ancestry in the UK Biobank. A linear mixed regression model was applied adjusting for age, sex, genotyping array, 10 PCs and genetic relatedness matrix, using BOLT-LMM with a MAF>0.001, BGEN imputation score >0.3, maximum per SNP missingness of 10%, and per sample missingness of 40% (Loh, Tucker et al. 2015). Reference 1000 genome European-ancestry (EUR) LD scores and genetic map (hg19) were implemented in this analysis. X-chromosome data were imputed and analyzed separately (with males coded as 0/2 and female genotypes coded as 0/1/2) using the same analytical approach in BOLT-LMM as was done for analysis of autosomes.”

PLINK: “Secondary GWAS excluding related individuals, shift workers, individuals who used psychiatric medications, and participants with chronic health and psychiatric illness (N=255,426) was performed adjusting for age, sex, genotyping array and 10 PCs in PLINK 1.9 (Chang, Chow et al. 2015). We used a hard-call genotype threshold of 0.1, SNP imputation quality threshold of 0.80, and a MAF threshold of 0.001. Gene-sex and gene-health status interaction analyses were performed on unrelated individuals using a linear regression model in PLINK with the additional --interaction flag.”

GCTA-COJO: “Conditional analyses to dissect independent signals in significant genomic regions were performed using GCTA-COJO (Yang, Ferreira et al. 2012) with MAF>0.001 and genome-wide significant threshold of $P < 5 \times 10^{-8}$ through a stepwise selection procedure using --cojo-slct flag.”

PICS: “Variant annotation for each significant locus was performed using PICS with 1000 Genome EUR LD reference with a causal probability of 0.2 or greater (Farh, Marson et al. 2015).”

3) Regional association plots for the 42 risk loci could be included as supplementary figures.

We added the regional association plots of genome-wide significant loci for self-reported daytime sleepiness (Supplementary Fig. 2) and for self-reported daytime sleepiness adjusted for BMI (Supplementary Fig. 4).

4) Numbers reported for the accelerometry-derived data differ: 85,388 (page 9, line 179), 85,499 (table 1) and 85,502 (page 20, line 434).

We thank the reviewer for this observation. We corrected the accelerometry data sample size to 85,388, which is the actual number use in this analysis.

5) Supplementary Table 2: For P=0, the exact values should be given.

We used R to perform Spearman correlation analysis. P-values smaller than machine epsilon (i.e., the relative error due to rounding in computer floating point arithmetic) were shown as 0. The minimum P-value in Supplementary Table 1 and 2 was $2.9e-317$. We cannot provide exact values for those P=0. We now added a footnote “P-values smaller than machine epsilon (the relative error due to rounding in computer floating point arithmetic) were shown as 0.”

6) Supplementary Table 4: The header reads “Replication in UK Biobank of genetic variants reported in prior studies to be associated with excessive daytime sleepiness and other related traits.” However, some of the loci in this table were not genome-wide significant in the previous GWAS, therefore this should not be called “replication”. A more neutral expression could be used such as “EDS GWAS association results of...”.

We changed the Supplementary Table 4 header as “Associations of previously reported genetic variants for daytime sleepiness and other related traits in the UK Biobank.”

7) Supplementary Table 5: typo, Sensitivity

We have corrected it as “Sensitivity”.

8) Supplementary Figure 6: typo?, (optional)

We have deleted “(Optional)”.

9) To avoid confusion, heritability estimates calculated from GWAS data should be referred to as “SNP-heritability”, e.g. page 11, line 231 or page 13, line 278.

We have changed “heritability” to “SNP-heritability” (Page 12, 14, 24), and also added its definition “defined as the proportion of trait variance explained by genome-wide additive genetic effects” (Page 24).

10) typo?: restless leg syndrome should be called restless legs syndrome, according to ICD-10, DSM-5.

We have corrected it to “restless legs syndrome”.

11) Supplementary Table 8 reports two genes identified in the sex-specific GWAS, but these results are not mentioned in the main text.

We added a sentence mentioning those genes in the main text as “Sex-stratified analyses on autosomes additionally identified *CWC27* and *DIAPH3* in women but not men; however, significant gene by sex interactions were not observed.” (Page 8)

12) For the main GWAS, self-reported EDS is used as the phenotype. This is important information and should be included in the title if possible.

As suggested by reviewer 2, we now changed the title to “Genome-wide association analysis of self-reported daytime sleepiness identifies 42 loci that suggest heterogeneous subgroups”.

13) Page 10, line 208: What makes the locus at rs13135092 pleiotropic? It was associated with self-reported sleep duration in a previous study, and with EDS in the current study. However, what evidence

is there for this being a pleiotropic effect and not EDS due to changes in sleep duration, which would reflect a shared pathway.

rs1315092 is pleiotropic because of its association with blood lipids, height, schizophrenia, and other traits in previous multi-phenotype association analysis. We added: “we further observed the pleiotropic locus at rs13135092 (*SLC39A8*, previously associated with blood lipids, height, schizophrenia, and other traits) (Pickrell, Berisa et al. 2016) to be significantly associated with bilateral putamen and striatum volume in the UK Biobank.” (Page 11)

Reviewer #2 (Remarks to the Author):

This manuscript reports on a genome-wide association study of daytime sleepiness. Despite some intriguing findings, there are several major concerns that need to be addressed.

1. The authors used the term “excessive daytime sleepiness (EDS)” in the manuscript. EDS generally indicates chronic and pathogenic sleepiness. The following question was utilized to confirm daytime sleepiness on page 18: “How likely are you to dose off or fall asleep during the daytime when you don’t mean to? (e.g. when working, reading or driving)” with the response options of “Never/rarely”, “sometimes”, “often”, “all of the time”. The question is not appropriate for ascertaining EDS. Epworth sleepiness scale (ESS), multiple sleep latency test (MSLT) and/or 24hr polysomnography (PSG) are needed to assess EDS. Therefore, I recommend that the term not “EDS” but “daytime sleepiness” should be used in this manuscript.

The title of this article should also include the term “self-reported”. It is important for sleep scientists to know whether subjective or objective measures were employed to assess daytime sleepiness when they check the title.

(“Genome-wide association analysis of self-reported daytime sleepiness identifies 42 loci that suggest phenotypic subgroups”)

We appreciate the reviewer’s suggestion. We changed the title to “Genome-wide association analyses of self-reported daytime sleepiness identifies 42 loci that suggest heterogeneous subgroups” and used “daytime sleepiness” in main text.

There is large inter-individual variability in levels of sleepiness (Van Dongen, Baynard et al. 2004, Kapur, Baldwin et al. 2005) and sleepiness can be assessed using subjective (questionnaire) or objective means (Mean Sleep Wakefulness Test or Multiple Sleep Latency Test [MSLT]). However, objective measurements are not feasible at large scale and are very sensitive to the prior night’s sleep. While only a single question was used in the UKB to assess sleepiness, it follows the framework of the ESS, which similarly asks about likelihood of dozing in several situations (in the ESS, each situation is identified as a separate item, while in the UKB question, the possible situations are clustered.) We describe this limitation in the discussion. “Primary analyses used self-reported daytime sleepiness expressed as a continuous variable derived from a 4-point scale. Future work should evaluate the psychometric properties of this question and compare it to other frequently used measures of daytime sleepiness, such as the Epworth Sleepiness Scale (ESS) or Maintenance of Wakefulness Test.” (Page17-18)

2. When sleep clinicians assess and diagnose EDS in patients, they have to consider insufficient sleep (Sleep. 2014; 37(6): 1035–1042.). Sleep scientists also need to treat insufficient sleep as an important confounding factor to appropriately obtain sleep data from subjects with pathogenic daytime sleepiness. Therefore, the author should discuss insufficient sleep in this manuscript. For example, “Did the UK biobank (UKB) collect data of insufficient sleep for the subjects?”, “How did insufficient sleep affect the results of this study?”, “Was it possible to control for the effect of insufficient sleep?” and so on.

We appreciate the reviewer’s concern regarding interpreting sleepiness in the context of sufficient/insufficient sleep. Unfortunately, this is a difficult task due to the wide range of sleep duration observed in individuals who do not report excessive sleepiness and the need to consider both sleep quality and sleep duration as determinants of “sufficient” sleep. We addressed this issue two ways:

A) We performed sensitivity analysis additionally adjusting for self-reported sleep duration and insomnia, and stratified analyses by short and long sleep duration to investigate the potential moderation or mediation that may relate to insufficient sleep. Those analyses did not substantially alter effect estimates for our identified signals (page 8).

B) We also performed hierarchical clustering analysis on sleepiness risk alleles with multi-night objective accelerometry-derived indices that capture sleep duration and continuity: sleep efficiency, sleep duration, sleep fragmentation, as well as with insomnia symptoms. These analyses identified two clusters of genetic loci. We interpret these analyses as showing that there are two clusters of genes that map to unique sets of actigraphy-based traits: one cluster likely relates to sleep propensity unrelated to insufficient sleep (and further supported by GTEx data showing enrichment in central nervous system areas) and the second cluster that likely partly reflects insufficient sleep (mapping to actigraphy measures of sleep fragmentation).

3. I was honestly a little surprised at results of replication studies in three cohorts. Only 8 loci out of 42 tested loci were replicated in the three cohorts. Probably, the 8 loci showed nominally significant associations because the Methods did not mention that a multiple testing correction such as bonferroni correction was applied for the replication studies (although 126 times were tested). In addition, effects of several SNPs, +/- of β values, were opposite between the UKB and replication studies in the Table S10. In my knowledge, genetic variants identified by self-reported chronotype GWASs were more replicated in independent studies (Nat Commun. 2016 9;7:10889, Supplementary Table 3.).

The authors described “Despite the challenges of individual loci replication with insufficient power, variable questionnaires across different cohorts and the multi-factorial etiology of EDS” on page 14 in the manuscript. I recognize the importance of sample sizes, however I consider that differences in questions between studies would affect the association results. Some readers might think that GWASs of daytime sleepiness are easily influenced by question content and question wording. The authors should more discuss this point because this is related to the reliability of the UKB daytime sleepiness questionnaire. For example, “How can differences in questionnaires be overcome?”, “Is the validation of the questionnaire using MSLT, PSG or ESS necessary in a future study” and so on.

We appreciate the reviewer’s concern about the validation and consistency of self-reported sleepiness questionnaires. As indicated in our response above, objective measurements are not feasible at large scale and are very sensitive to the prior night’s sleep. While only a single question was used in the UKB

to assess sleepiness, the question used follows the framework of the ESS, which similarly asks about likelihood of dozing in several situations (in the ESS each situation is identified as a separate item, while in the UKB question, the possible situations are clustered). While we agree that the psychometric properties of this question should be studied, it is important to highlight that in addition to the UKB sleepiness question, we analyzed objectively estimates sleep-wake indices from actigraphy in a subset of the UK Biobank sample. (Page 9; Table S11). We add to the discussion: “Primary analyses used self-reported daytime sleepiness expressed as a continuous variable derived from a 4-point like scale. Future work should evaluate the psychometric properties of this question and compare it to other frequently used measures of daytime sleepiness, such as the Epworth Sleepiness Scale (ESS) or Maintenance of Wakefulness Test. It is likely that there was some loss of power due to use of a single measure of self-reported sleepiness resulting in random misclassification. However, the large sample for which questionnaire data were available provided results that were able to be further studied in a smaller sample of 7-day accelerometry-derived sleep data (which has been shown to agree well with polysomnography). Future work using objective measurements of sleepiness, such as from vigilance tests, may provide further insights into the genetics of sleepiness-related traits.” (Page17-18)

4. I consider that rs7598712 was not replicated in the Finrisk although the p value was 0.045 in the Table S10. The reason is that the β value of the UKB was 0.006, while that of the Finrisk was -0.012. These effects seem to be opposite.

Thanks for pointing out the inconsistent direction. We have updated this analysis and now five individual signals were marginally replicated with $P < 0.05$ and consistent association direction in two replication datasets with sleepiness measures. (Page 8)

5. The authors performed a linear mixed regression model using EDS as a continuous variable of 4 integers (on Page 22). The number of subjects who answered “all of the time” was only 29. Are there possibilities that the small sample size causes problems for your analyses? Is it reasonable for regression analyses to bring two groups, “often” and “all of the time”, into one category?

We treated the self-reported daytime sleepiness as a continuous variable derived from a 4-point scale as was done in the GWAS using UKB phase I analysis. We now tried another analysis grouping “often” and “all of the time” together and testing the association with 42 loci in unrelated individuals using PLINK. The results are almost identical (following Table).

Table R2. Associations between 42 loci with self-reported sleepiness using different coding in unrelated individuals.						
SNP	pos	Nearest Gene(s)	4-value continuous sleepiness		3-value continuous sleepiness	
			BETA (SE)	P	BETA (SE)	P
rs2787120	1:33306297	S100PBP	-0.009 (0.002)	9.70E-09	-0.009 (0.002)	9.27E-09
rs12140153	1:62579891	PATJ	-0.018 (0.002)	5.45E-17	-0.018 (0.002)	6.04E-17
rs17131124	1:91127548	ZNF326, BARHL2	0.012 (0.002)	9.66E-09	0.012 (0.002)	9.61E-09
rs57746981	1:201885234	LMOD1	-0.006 (0.001)	8.19E-06	-0.006 (0.001)	7.81E-06
rs825127	1:223506788	SUSD4	-0.006 (0.001)	6.69E-08	-0.006 (0.001)	7.10E-08
rs4665972	2:27598097	SNX17	0.007 (0.001)	8.19E-09	0.007 (0.001)	8.69E-09
rs7598712	2:46660452	TMEM247	-0.006 (0.001)	1.44E-07	-0.006 (0.001)	1.48E-07
rs6741951	2:58959112	LOC644456, LOC730134	-0.006 (0.001)	9.90E-06	-0.006 (0.001)	1.01E-05
rs11123962	2:104157011	LOC728815, LOC644265	0.008 (0.001)	5.95E-11	0.008 (0.001)	5.47E-11
rs9712275	2:198907143	PLCL1	-0.006 (0.001)	7.18E-07	-0.006 (0.001)	7.93E-07
rs7607363	2:213402705	ERBB4	0.004 (0.001)	2.86E-04	0.004 (0.001)	3.10E-04
rs13010456	2:236792801	AGAP1	-0.009 (0.001)	9.86E-14	-0.009 (0.001)	8.42E-14

rs13097760	3:82823561	GBE1, CYP51P1	0.006 (0.001)	3.52E-06	0.006 (0.001)	3.68E-06
rs9875075	3:84563848	CYP51P1, LOC100131101	0.006 (0.001)	9.77E-06	0.006 (0.001)	8.71E-06
rs960986	3:85519305	LOC440970, CADM2	-0.007 (0.001)	4.93E-08	-0.007 (0.001)	5.13E-08
rs843372	3:183996213	ECEF2	0.008 (0.001)	3.44E-08	0.008 (0.001)	3.72E-08
rs11942333	4:46389486	GABRA2	0.006 (0.001)	5.51E-06	0.006 (0.001)	6.46E-06
rs13135092	4:103198082	SLC39A8	0.01 (0.002)	7.96E-06	0.01 (0.002)	1.03E-05
rs6897863	5:92512481	CCT7P2, LOC391811	-0.007 (0.001)	2.61E-09	-0.007 (0.001)	2.66E-09
rs12153518	5:138501494	SIL1	0.007 (0.001)	1.66E-08	0.007 (0.001)	1.84E-08
rs6923811	6:27289776	POM121L2, FKSG83	-0.007 (0.001)	2.93E-08	-0.007 (0.001)	2.48E-08
rs55960940	6:38153146	BTBD9	-0.007 (0.002)	4.82E-06	-0.007 (0.002)	4.89E-06
rs3122170	6:55058998	HCRTR2	0.009 (0.001)	2.55E-10	0.009 (0.001)	2.52E-10
rs62519825	8:65479707	LOC100129963	0.009 (0.002)	1.04E-06	0.009 (0.002)	9.80E-07
rs285793	8:106087862	LOC644103, LOC100128132	0.007 (0.001)	5.31E-09	0.007 (0.001)	5.98E-09
rs7837226	8:131235895	ASAP1	-0.006 (0.001)	2.35E-07	-0.006 (0.001)	2.49E-07
rs55818482	9:81744922	KRT18P24, CHCHD9	0.01 (0.001)	3.12E-12	0.01 (0.001)	3.11E-12
rs1566362	9:128163132	GAPVD1, MAPKAP1	-0.007 (0.001)	2.59E-08	-0.007 (0.001)	2.57E-08
rs7476897	10:92416402	LOC119358, HTR7	-0.007 (0.001)	5.56E-09	-0.007 (0.001)	5.12E-09
rs4765939	12:2582397	CACNA1C	0.005 (0.001)	1.62E-04	0.005 (0.001)	1.59E-04
rs1846644	12:117938380	KSR2	0.011 (0.001)	2.52E-18	0.011 (0.001)	1.83E-18
rs8015449	14:82161860	EEF1A1P2, RPL9P6	-0.005 (0.001)	6.78E-06	-0.005 (0.001)	7.06E-06
rs17356118	15:83237899	CPEB1	0.008 (0.001)	1.94E-08	0.008 (0.001)	1.60E-08
rs886114	16:23865986	PRKCB	0.005 (0.001)	2.06E-04	0.005 (0.001)	2.15E-04
rs11649804	17:17696755	RAI1	-0.006 (0.001)	2.82E-06	-0.006 (0.001)	2.69E-06
rs62055936	17:43848761	LOC644191, MGC57346	-0.009 (0.001)	1.83E-09	-0.009 (0.001)	1.71E-09
rs2048522	18:44800515	FUSSEL18, TPMTP1	-0.006 (0.001)	1.07E-06	-0.006 (0.001)	1.01E-06
rs641498	6:124911565	NKAIN2	-0.007 (0.001)	4.13E-08	-0.007 (0.001)	4.71E-08
rs12253139	10:128899947	DOCK1	0.008 (0.002)	2.26E-05	0.008 (0.002)	2.21E-05
rs7982022	13:54049003	LOC100133285, RP11-365K22.1	0.006 (0.001)	4.25E-06	0.006 (0.001)	4.34E-06
rs2472297	15:75027880	CYP1A1, CYP1A2	-0.006 (0.001)	5.77E-06	-0.006 (0.001)	5.09E-06
rs7162082	15:83896608	HDGFRP3, BNC1	-0.006 (0.001)	8.99E-05	-0.006 (0.001)	9.14E-05

6. In the Table S14, more genes were identified by a gene-based association analysis, compared to 42 loci. Associated genetic variants in each significant gene were not included in the Table S14. Could the authors give us the information of the genetic variants with annotation?

(Were there so many SNPs? Did pascal compute only gene scores and gene p-values?)

The PASCAL software constructs an association score from the summary statistics of multiple SNPs (significant or not) in a gene and estimates gene-based association p-values. It provides independent evidence of gene-based associations with sleepiness. No single variants were reported in this case in Supplementary Table 14. In Supplementary Table 12, we reported 164 genes within the association interval of each locus (SNPs with $r^2 \geq 0.5$ to lead SNP). PASCAL identified 61 genes that overlapped with Supplementary Table 12, in addition to another 33 genes. We now added a sentence to explain the analysis of Pascal in methods “Gene-based analysis was performed using PASCAL, which estimated a combined association p-value from the summary statistics of multiple SNPs in a gene (Lamparter, Marbach et al. 2016)”. (Page 26).

7. I do not understand the following sentence on page 11: “Gene-based analyses using PASCAL identified 94 genes associated with EDS ($p < 2.29 \times 10^{-6}$) (Supplementary Table 14), of which 63 overlapped with genes under significant association peaks shown in Supplementary Table 13.” I could not find the 63 genes.

We apologize for mistakenly referring to the wrong table and have now corrected this error. We double checked the overlapped genes and changed the sentence to “..., of which 61 overlapped with genes under significant association peaks shown in supplementary Table 12”. (Page 12).

References

- Carmelli, D., D. L. Bliwise, G. E. Swan and T. Reed (2001). "A genetic analysis of the Epworth Sleepiness Scale in 1560 World War II male veteran twins in the NAS-NRC Twin Registry." *J Sleep Res* **10**(1): 53-58.
- Chang, C. C., C. C. Chow, L. C. Tellier, S. Vattikuti, S. M. Purcell and J. J. Lee (2015). "Second-generation PLINK: rising to the challenge of larger and richer datasets." *Gigascience* **4**: 7.
- Cotto, K. C., A. H. Wagner, Y. Y. Feng, S. Kiwala, A. C. Coffman, G. Spies, A. Wollam, N. C. Spies, O. L. Griffith and M. Griffith (2018). "DGIdb 3.0: a redesign and expansion of the drug-gene interaction database." *Nucleic Acids Res* **46**(D1): D1068-D1073.
- Desai, A. V., L. F. Cherkas, T. D. Spector and A. J. Williams (2004). "Genetic influences in self-reported symptoms of obstructive sleep apnoea and restless legs: a twin study." *Twin Res* **7**(6): 589-595.
- Farh, K. K., A. Marson, J. Zhu, M. Kleinewietfeld, W. J. Housley, S. Beik, N. Shores, H. Whitton, R. J. Ryan, A. A. Shishkin, M. Hatan, M. J. Carrasco-Alfonso, D. Mayer, C. J. Luckey, N. A. Patsopoulos, P. L. De Jager, V. K. Kuchroo, C. B. Epstein, M. J. Daly, D. A. Hafler and B. E. Bernstein (2015). "Genetic and epigenetic fine mapping of causal autoimmune disease variants." *Nature* **518**(7539): 337-343.
- Hamosh, A., A. F. Scott, J. S. Amberger, C. A. Bocchini and V. A. McKusick (2005). "Online Mendelian Inheritance in Man (OMIM), a knowledgebase of human genes and genetic disorders." *Nucleic Acids Res* **33**(Database issue): D514-517.
- Kapur, V. K., C. M. Baldwin, H. E. Resnick, D. J. Gottlieb and F. J. Nieto (2005). "Sleepiness in patients with moderate to severe sleep-disordered breathing." *Sleep* **28**(4): 472-477.
- Lamparter, D., D. Marbach, R. Rueedi, Z. Kutalik and S. Bergmann (2016). "Fast and Rigorous Computation of Gene and Pathway Scores from SNP-Based Summary Statistics." *PLoS Comput Biol* **12**(1): e1004714.
- Lane, J. M., J. Liang, I. Vlasac, S. G. Anderson, D. A. Bechtold, J. Bowden, R. Emsley, S. Gill, M. A. Little, A. I. Luik, A. Loudon, F. A. Scheer, S. M. Purcell, S. D. Kyle, D. A. Lawlor, X. Zhu, S. Redline, D. W. Ray, M. K. Rutter and R. Saxena (2017). "Genome-wide association analyses of sleep disturbance traits identify new loci and highlight shared genetics with neuropsychiatric and metabolic traits." *Nat Genet* **49**(2): 274-281.
- Loh, P. R., G. Tucker, B. K. Bulik-Sullivan, B. J. Vilhjalmsson, H. K. Finucane, R. M. Salem, D. I. Chasman, P. M. Ridker, B. M. Neale, B. Berger, N. Patterson and A. L. Price (2015). "Efficient Bayesian mixed-model analysis increases association power in large cohorts." *Nat Genet* **47**(3): 284-290.
- MacArthur, J., E. Bowler, M. Cerezo, L. Gil, P. Hall, E. Hastings, H. Junkins, A. McMahon, A. Milano, J. Morales, Z. M. Pendlington, D. Welter, T. Burdett, L. Hindorff, P. Flicek, F. Cunningham and H. Parkinson (2017). "The new NHGRI-EBI Catalog of published genome-wide association studies (GWAS Catalog)." *Nucleic Acids Res* **45**(D1): D896-D901.
- Pickrell, J. K., T. Berisa, J. Z. Liu, L. Segurel, J. Y. Tung and D. A. Hinds (2016). "Detection and interpretation of shared genetic influences on 42 human traits." *Nat Genet* **48**(7): 709-717.
- Ramos, E. M., D. Hoffman, H. A. Junkins, D. Maglott, L. Phan, S. T. Sherry, M. Feolo and L. A. Hindorff (2014). "Phenotype-Genotype Integrator (PheGenI): synthesizing genome-wide association study (GWAS) data with existing genomic resources." *Eur J Hum Genet* **22**(1): 144-147.
- Slowikowski, K., X. Hu and S. Raychaudhuri (2014). "SNPsea: an algorithm to identify cell types, tissues and pathways affected by risk loci." *Bioinformatics* **30**(17): 2496-2497.
- Van Dongen, H. P., M. D. Baynard, G. Maislin and D. F. Dinges (2004). "Systematic interindividual differences in neurobehavioral impairment from sleep loss: evidence of trait-like differential vulnerability." *Sleep* **27**(3): 423-433.
- Van Dongen, H. P., K. M. Vitellaro and D. F. Dinges (2005). "Individual differences in adult human sleep and wakefulness: Leitmotif for a research agenda." *Sleep* **28**(4): 479-496.

van Hees, V. T., S. Sabia, S. E. Jones, A. R. Wood, K. N. Anderson, M. Kivimaki, T. M. Frayling, A. I. Pack, M. Bucan, M. I. Trenell, D. R. Mazzotti, P. R. Gehrman, B. A. Singh-Manoux and M. N. Weedon (2018). "Estimating sleep parameters using an accelerometer without sleep diary." *Sci Rep* **8**(1): 12975.

Wain, L. V., N. Shrine, S. Miller, V. E. Jackson, I. Ntalla, M. Soler Artigas, C. K. Billington, A. K. Kheirallah, R. Allen, J. P. Cook, K. Probert, M. Obeidat, Y. Bosse, K. Hao, D. S. Postma, P. D. Pare, A. Ramasamy, U. K. B. E. Consortium, R. Magi, E. Mihailov, E. Reinmaa, E. Melen, J. O'Connell, E. Frangou, O. Delaneau, G. S. K. C. Ox, C. Freeman, D. Petkova, M. McCarthy, I. Sayers, P. Deloukas, R. Hubbard, I. Pavord, A. L. Hansell, N. C. Thomson, E. Zeggini, A. P. Morris, J. Marchini, D. P. Strachan, M. D. Tobin and I. P. Hall (2015). "Novel insights into the genetics of smoking behaviour, lung function, and chronic obstructive pulmonary disease (UK BiLEVE): a genetic association study in UK Biobank." *Lancet Respir Med* **3**(10): 769-781.

Watson, N. F., J. Goldberg, L. Arguelles and D. Buchwald (2006). "Genetic and environmental influences on insomnia, daytime sleepiness, and obesity in twins." *Sleep* **29**(5): 645-649.

Yang, J., T. Ferreira, A. P. Morris, S. E. Medland, A. T. C. Genetic Investigation of, D. I. G. Replication, C. Meta-analysis, P. A. Madden, A. C. Heath, N. G. Martin, G. W. Montgomery, M. N. Weedon, R. J. Loos, T. M. Frayling, M. I. McCarthy, J. N. Hirschhorn, M. E. Goddard and P. M. Visscher (2012). "Conditional and joint multiple-SNP analysis of GWAS summary statistics identifies additional variants influencing complex traits." *Nat Genet* **44**(4): 369-375, S361-363.

Reviewers' Comments:

Reviewer #1:

Remarks to the Author:

The authors have addressed most of my comments to my satisfaction. However, a few concerns remain that I would like the authors to consider:

1. Use of the term "subgroup". I still get confused when reading the term subgroup in relation to the clustering of sleepiness loci to the two biologically defined entities "sleep propensity" and "sleep fragmentation". In the discussion, the authors also use the expression "phenotypic subtypes" (page 16). It is likely that many readers will connect the terms subgroup and phenotypic subtypes to the clinical presentation of sleepiness/EDS, especially as the manuscript also raises the issue of heterogeneity in EDS phenotypes captured by the different methods used for assessing sleepiness in the study cohorts.

If there were EDS phenotypic subtypes, I would expect an analysis where the authors show that there are subgroups of individuals with e.g. severe sleepiness who show a stronger association for a GRS built from loci for sleep fragmentation than for the GRS of loci for sleep propensity.

To avoid confusion, I would suggest to clarify by using "biological subtypes" or "subtypes of biological processes/mechanisms".

2. Replication of risk loci in additional cohorts: In the discussion, the authors state that five loci were replicated (page 15). Considering the fact that these loci show only nominal significance (no P values corrected for multiple testing are reported) and that, despite consistent direction of effects, the effect sizes are quite different between the cohorts, this statement seems to strong. The GRS seems to be driven by three loci with nominal significance in the meta-analysis as there is a rather large drop in significance when excluding those three loci. What is the difference in effect size of these two GRS? This should be addressed in the discussion. The heterogeneity in effect sizes and possible explanations should also be discussed in more detail.

3. Sensitivity analyses: What is the reason for the difference in association strength seen for some loci, comparing e.g. Supplementary Table 3 results to Supplementary Table 5 and 6? Only sample size? Some Loci with same effect sizes estimates have quite different results: one has almost the same association P value in both analyses, whereas another one shows quite different association strength.

4. SNPs reported for association analyses in Supplementary Tables are inconsistent for some loci: When comparing UKB-based results for analyses, differing lead SNPs are given for three loci in Supplementary tables 3, 5, 6, and 10:

Locus CYP51P1, LOC100131101: rs9875075 (Tables 3 and 10) and rs34478464 (Tables 5 and 6)

Locus RAI1: rs11649804 (Tables 3 and 10) and rs11078398 (Tables 5 and 6)

Locus LOC644191, MGC57346: rs62055936 (Tables 3 and 10) and rs62066119 (Tables 5 and 6)

5. Minor concern: QQ-plots in the supplement show lambda values of around 1.14: Is this in line with other studies using UKB data? Is this due to polygenicity? The authors could add the intercept results from their LD-score regression analysis to clarify this.

Reviewer #2:

Remarks to the Author:

The revised manuscript, NCOMMS-18-34519A, warrants publication in Nature Communications from my point of view. Authors have addressed all points from the reviewer.

The authors have addressed most of my comments to my satisfaction. However, a few concerns remain that I would like the authors to consider:

1. Use of the term “subgroup”. I still get confused when reading the term subgroup in relation to the clustering of sleepiness loci to the two biologically defined entities “sleep propensity” and “sleep fragmentation”. In the discussion, the authors also use the expression “phenotypic subtypes” (page 16). It is likely that many readers will connect the terms subgroup and phenotypic subtypes to the clinical presentation of sleepiness/EDS, especially as the manuscript also raises the issue of heterogeneity in EDS phenotypes captured by the different methods used for assessing sleepiness in the study cohorts. If there were EDS phenotypic subtypes, I would expect an analysis where the authors show that there are subgroups of individuals with e.g. severe sleepiness who show a stronger association for a GRS built from loci for sleep fragmentation than for the GRS of loci for sleep propensity. To avoid confusion, I would suggest to clarify by using “biological subtypes” or “subtypes of biological processes/mechanisms”.

Thanks for the reviewer’s suggestion. We have now changed the term to “biological subtypes” (Title, Abstract, Page 9, Page 16, Page 17, Table 1).

2. Replication of risk loci in additional cohorts: In the discussion, the authors state that five loci were replicated (page 15). Considering the fact that these loci show only nominal significance (no P values corrected for multiple testing are reported) and that, despite consistent direction of effects, the effect sizes are quite different between the cohorts, this statement seems to strong.

We have qualified this statement in response to the reviewer.

We cannot compare the effect sizes across different cohorts because of the different scales of sleepiness. In UKB, sleepiness was coded in a 4-point scale (“never”, “sometimes”, “often”, or “all of the time”). In HUNT, sleepiness was coded in a 3-point scale (“Never/seldom”, “Sometimes”, and “Several times”). In Health 2000, sleepiness was measured using Epworth Sleepiness Scale (0-24). However, we still observed nominal significance and consistent directions of effects.

We have now added the scale differences and “nominal” in the discussion (Page 15) “Despite the challenges of individual loci replication with insufficient power (5-57% in replication cohorts; Supplementary Table 10), variable questionnaires in different scales across different cohorts and the multi-factorial etiology of sleepiness, we observed nominal replication at five loci...”

The GRS seems to be driven by three loci with nominal significance in the meta-analysis as there is a rather large drop in significance when excluding those three loci. What is the difference in effect size of these two GRS? This should be addressed in the discussion. The heterogeneity in effect sizes and possible explanations should also be discussed in more detail.

As shown in the table below, the effect estimates dropped modestly in HUNT (32%) and Health 2000 (14%) excluding the three loci. The p-value of the meta-analysis remains significant. This observation suggests that in addition to the three nominal significant loci, there are additional individual sleepiness loci with small effects that contribute to the combined effect of GRS. We have now added in discussion

“While the GRS association was highly significant including the three loci with nominal significance in the meta-analysis, the effect estimates removing the three loci remained at 67% of the original effect in HUNT and 86% of the original effect in Health 2000, suggesting that additional individual sleepiness loci contribute to the combined effect of the GRS. However, replication in additional, well-powered cohorts will be important”.

Replication of Sleepiness GRS in HUNT and Health 2000.							
	HUNT sleepiness (N=29,906)			Health 2000 ESS (N=4,546)			Meta-analysis of Sleepiness
	β	se	P-value	β	se	P-value	Fisher's P-value
42 Sleepiness loci	0.361	0.118	0.002	4.045	1.631	0.013	3.00E-04
Excluding 3 loci	0.242	0.126	0.054	3.48	1.742	0.046	0.017

3. Sensitivity analyses: What is the reason for the difference in association strength seen for some loci, comparing e.g. Supplementary Table 3 results to Supplementary Table 5 and 6? Only sample size? Some Loci with same effect sizes estimates have quite different results: one has almost the same association P value in both analyses, whereas another one shows quite different association strength.

There are two major differences between the primary GWAS analyses results presented in Supplementary Table 3 and the sensitivity analyses at baseline in Supplementary Table 5 and 6 performed to compare this baseline against covariate-adjusted estimates or in stratified analyses. 1) Sample size: The former analysis was performed on all available individuals (N=452,071); the latter was performed on unrelated individuals (N=337,539). 2) Software: the former used BOLT-LMM adjusting for PCs and genetic relatedness matrix to correct for population and relatedness structure; the latter used PLINK adjusting for PCs to correct for population structure. There could be a number of reasons for observing different association strengths at some loci, including power based on sample size, the distribution differences of the genotypes in each sleepiness category, potential uncorrected confounders (e.g. long-distance relatedness in PLINK analysis), or could just be chance differences. We cannot speculate the reason given the number of associations and potential reasons.

4. SNPs reported for association analyses in Supplementary Tables are inconsistent for some loci: When comparing UKB-based results for analyses, differing lead SNPs are given for three loci in Supplementary tables 3, 5, 6, and 10:

Locus CYP51P1, LOC100131101: rs9875075 (Tables 3 and 10) and rs34478464 (Tables 5 and 6)

Locus RAI1: rs11649804 (Tables 3 and 10) and rs11078398 (Tables 5 and 6)

Locus LOC644191, MGC57346: rs62055936 (Tables 3 and 10) and rs62066119 (Tables 5 and 6)

We apologize for the confusion. Since we used different SNP filter threshold for the BOLT-LMM and PLINK, those three leading SNPs were not available in the PLINK analysis. Proxy SNPs in complete LD ($r^2=1$) with the leading SNPs in 1000G EUR reference were chosen instead. We have now added this information in the footnote.

5. Minor concern: QQ-plots in the supplement show lambda values of around 1.14: Is this in line with other studies using UKB data? Is this due to polygenicity? The authors could add the intercept results from their LD-score regression analysis to clarify this.

The lambda departure of 1.14 are common in UKB studies due to the polygenicity of complex traits. We have calculated the LD-score regression intercept=1.0179. We have now added that value on Supplementary Figures.

GC: Genomic Control; LDSR: LD Score Regression.

Comparison between BOLT-LMM results and Sensitivity baseline results.													
		BOLT-LMM results in 450,221 individuals					PLINK results in 337,539 individuals						
SNP	Pos	Alleles (E/A)	EAf	β	se	P	SNP	pos	Alleles (E/A)	EAf	β	se	P
rs2787120	1:33306297	A/G	0.8326	0.008	0.001	2.00E-08	rs2787120	1:33306297	A/G	0.8325	0.009	0.002	9.70E-09
rs12140153	1:62579891	G/T	0.5631	0.017	0.002	2.80E-20	rs12140153	1:62579891	G/T	0.9102	0.018	0.002	5.45E-17
rs17131124	1:91127548	C/G	0.9115	-0.011	0.002	1.70E-09	rs17131124	1:91127548	C/G	0.9162	-0.012	0.002	9.66E-09
rs57746981	1:201885234	C/T	0.6445	0.007	0.001	2.20E-10	rs57746981	1:201885234	C/T	0.6444	0.006	0.001	8.19E-06
rs825127	1:223506788	T/G	0.5308	0.006	0.001	9.50E-09	rs825127	1:223506788	T/G	0.5303	0.006	0.001	6.69E-08
rs4665972	2:27598097	T/C	0.3936	0.007	0.001	3.90E-10	rs4665972	2:27598097	T/C	0.3920	0.007	0.001	8.19E-09
rs7598712	2:46660452	G/T	0.5553	0.006	0.001	2.20E-08	rs7598712	2:46660452	G/T	0.5560	0.006	0.001	1.44E-07
rs6741951	2:58959112	G/A	0.7108	0.007	0.001	2.70E-09	rs6741951	2:58959112	G/A	0.7127	0.006	0.001	9.90E-06
rs11123962	2:104157011	T/G	0.5532	-0.008	0.001	7.50E-15	rs11123962	2:104157011	T/G	0.5518	-0.008	0.001	5.95E-11
rs9712275	2:198907143	C/T	0.4859	-0.006	0.001	1.30E-08	rs9712275	2:198907143	C/T	0.4861	-0.006	0.001	7.18E-07
rs7607363	2:213402705	A/G	0.5612	-0.006	0.001	8.00E-09	rs7607363	2:213402705	A/G	0.5608	-0.004	0.001	2.86E-04
rs13010456	2:236792801	A/G	0.5947	0.008	0.001	2.10E-13	rs13010456	2:236792801	A/G	0.5954	0.009	0.001	9.86E-14
rs13097760	3:82823561	A/C	0.6387	-0.006	0.001	3.20E-08	rs13097760	3:82823561	A/C	0.6408	-0.006	0.001	3.52E-06
rs34478464*	3:84228726	C/T	0.8081	-0.009	0.001	5.80E-11	rs9875075	3:84563848	A/G	0.7292	-0.006	0.001	9.77E-06
rs960986	3:85519305	C/T	0.6365	0.007	0.001	1.50E-11	rs960986	3:85519305	C/T	0.6356	0.007	0.001	4.93E-08
rs843372	3:183996213	C/T	0.2301	0.008	0.001	2.20E-11	rs843372	3:183996213	C/T	0.2293	0.008	0.001	3.44E-08
rs11942333	4:46389486	G/A	0.6758	-0.006	0.001	3.80E-08	rs11942333	4:46389486	G/A	0.6757	-0.006	0.001	5.51E-06
rs13135092	4:103198082	A/G	0.9172	-0.010	0.002	3.10E-08	rs13135092	4:103198082	A/G	0.9182	-0.01	0.002	7.96E-06
rs6897863	5:92512481	A/C	0.5844	0.006	0.001	7.60E-10	rs6897863	5:92512481	A/C	0.5833	0.007	0.001	2.61E-09
rs12153518	5:138501494	A/C	0.4722	0.007	0.001	6.80E-11	rs12153518	5:138501494	A/C	0.4721	0.007	0.001	1.66E-08
rs6923811	6:27289776	T/C	0.6790	0.007	0.001	9.10E-10	rs6923811	6:27289776	T/C	0.6789	0.007	0.001	2.93E-08
rs55960940	6:38153146	T/C	0.8222	0.008	0.001	2.00E-08	rs55960940	6:38153146	T/C	0.8230	0.007	0.002	4.82E-06
rs3122170	6:55058998	C/A	0.2311	0.010	0.001	5.60E-15	rs3122170	6:55058998	C/A	0.2299	0.009	0.001	2.55E-10
rs62519825	8:65479707	T/C	0.8874	-0.009	0.002	3.80E-09	rs62519825	8:65479707	T/C	0.8868	-0.009	0.002	1.04E-06

rs285793	8:106087862	G/A	0.4612	0.007	0.001	7.90E-11	rs285793	8:106087862	G/A	0.4619	0.007	0.001	5.31E-09
rs7837226	8:131235895	A/G	0.4729	-0.006	0.001	2.00E-08	rs7837226	8:131235895	A/G	0.4732	-0.006	0.001	2.35E-07
rs55818482	9:81744922	T/C	0.7850	-0.010	0.001	1.40E-14	rs55818482	9:81744922	T/C	0.7874	-0.01	0.001	3.12E-12
rs1566362	9:128163132	T/C	0.6317	0.006	0.001	3.80E-09	rs1566362	9:128163132	T/C	0.6319	0.007	0.001	2.59E-08
rs7476897	10:92416402	G/A	0.6794	0.007	0.001	2.70E-11	rs7476897	10:92416402	G/A	0.6795	0.007	0.001	5.56E-09
rs4765939	12:2582397	G/C	0.5833	-0.006	0.001	2.00E-09	rs4765939	12:2582397	G/C	0.5845	-0.005	0.001	1.62E-04
rs1846644	12:117938380	T/C	0.5908	-0.011	0.001	2.50E-27	rs1846644	12:117938380	T/C	0.5894	-0.011	0.001	2.52E-18
rs8015449	14:82161860	A/G	0.5386	0.006	0.001	1.90E-09	rs8015449	14:82161860	A/G	0.5387	0.005	0.001	6.78E-06
rs17356118	15:83237899	A/G	0.7688	-0.008	0.001	2.60E-10	rs17356118	15:83237899	A/G	0.7697	-0.008	0.001	1.94E-08
rs886114	16:23865986	C/T	0.3573	0.006	0.001	1.90E-08	rs886114	16:23865986	C/T	0.3555	0.005	0.001	2.06E-04
rs11078398*	17:17697099	G/A	0.7437	0.008	0.001	7.10E-10	rs11649804	17:17696755	C/A	0.7109	0.006	0.001	2.82E-06
rs62066119*	17:43695197	C/T	0.7479	0.008	0.001	7.30E-12	rs62055936	17:43848761	T/A	0.7765	0.009	0.001	1.83E-09
rs2048522	18:44800515	A/T	0.5654	0.006	0.001	3.50E-08	rs2048522	18:44800515	A/T	0.5664	0.006	0.001	1.07E-06
rs641498	6:124911565	A/G	0.39	-0.005	0.001	2.70E-07	rs641498	6:124911565	A/G	0.391	-0.007	0.001	4.13E-08
rs12253139	10:128899947	T/C	0.88	-0.008	0.002	1.20E-07	rs12253139	10:128899947	T/C	0.877	-0.008	0.002	2.26E-05
rs7982022	13:54049003	G/A	0.56	-0.006	0.001	1.60E-07	rs7982022	13:54049003	G/A	0.566	-0.006	0.001	4.25E-06
rs2472297	15:75027880	C/T	0.74	0.006	0.001	6.80E-07	rs2472297	15:75027880	C/T	0.733	0.006	0.001	5.77E-06
rs7162082	15:83896608	C/T	0.8	0.007	0.001	6.60E-08	rs7162082	15:83896608	C/T	0.795	0.006	0.001	8.99E-05

*For SNPs reported in BOLT analysis (Supplementary Table 3) but not available in PLINK analysis because of filtering criteria differences, proxy SNPs with complete LD ($r^2=1$) in 1000 Genome EUR reference panel was chosen for sensitivity analysis.

Reviewers' Comments:

Reviewer #1:

Remarks to the Author:

The authors have addressed my concerns, the revised manuscript is suitable for publication.